# Analytic and numerical bootstrap of CFTs with $O(m) \times O(n)$ global symmetry in 3D

**Johan Henriksson[1,2], Stefanos R. Kousvos[3,4] and Andreas Stergiou[5]**

**1** Mathematical Institute, University of Oxford, Andrew Wiles Building,
Radcliffe Observatory, Quarter, Woodstock Road, Oxford, OX2 6GG, UK
**2** Lincoln College, University of Oxford, Turl Street, Oxford, OX1 3DR, UK
**3** Department of Physics, University of Crete, Heraklion, GR-70013, Greece
**4** Institute of Theoretical and Computational Physics (ITCP),
Department of Physics, University of Crete, 70013 Heraklion, Greece
**5** Theoretical Division, MS B285, Los Alamos National Laboratory,
Los Alamos, NM 87545, USA

## Abstract

Motivated by applications to critical phenomena and open theoretical questions, we study conformal field theories with $O(m) \times O(n)$ global symmetry in $d = 3$ spacetime dimensions. We use both analytic and numerical bootstrap techniques. Using the analytic bootstrap, we calculate anomalous dimensions and OPE coefficients as power series in $\varepsilon = 4 - d$ and in $1/n$, with a method that generalizes to arbitrary global symmetry. Whenever comparison is possible, our results agree with earlier results obtained with diagrammatic methods in the literature. Using the numerical bootstrap, we obtain a wide variety of operator dimension bounds, and we find several islands (isolated allowed regions) in parameter space for $O(2) \times O(n)$ theories for various values of $n$. Some of these islands can be attributed to fixed points predicted by perturbative methods like the $\varepsilon$ and large-$n$ expansions, while others appear to arise due to fixed points that have been claimed to exist in resummations of perturbative beta functions.



# 1 Introduction

The Landau theory of phase transitions [1] has provided a powerful framework for the search of emergent critical behavior for decades. It has served as a solid foundation for the development of renormalization group (RG) methods like the $\varepsilon$ expansion [2,3], Monte Carlo [4] and functional RG [5,6]. These methods have been very successful in predicting critical behavior in a wide variety of situations, but there is still a surprising number of discrepancies and disagreements in the literature, pointing to potentially deep underlying physical principles. More recently, following pioneering work of [7–9] and especially [10], old conformal bootstrap ideas have morphed into an an entirely new and computationally rigorous approach for the study of conformal field theories (CFTs)—for a review see [11] and for an introduction [12].

In this work we use the conformal bootstrap method, both analytically and numerically, to study CFTs with $O(m) \times O(n)$ symmetry. The importance and relevance of this undertaking is highlighted both by the experimental applications of such CFTs, as well as their inherent theoretical interest. CFTs with symmetry $O(2) \times O(2)$ and $O(2) \times O(3)$ should describe second order phase transitions in a wide variety of materials [13, 14], and indeed such transitions have been claimed to be observed in various experiments. Our work is also of pure theoretical interest, since it attempts to shed light on the possible existence of fixed points that arise due to resummations of perturbative beta functions. The existence of such fixed points has been questioned by some functional RG studies [15–17]. Thus, theoretically the state of affairs regarding these fixed points has remained murky despite decades of effort, with conflicting results obtained by different RG methods.

The most unequivocal results for $O(m) \times O(n)$ CFTs have been obtained by taking $n$ large and $m$ finite. The existence of a well-defined large-$n$ expansion was established in [18], and

the strongest results to date have appeared in [19–21]. The $\varepsilon$ expansion has been widely used as well, see e.g. [13,14,18,22–26], with the highest-loop study (six loops) performed recently in [27].[1] Here we will use the analytic bootstrap method of large spin perturbation theory, introduced in [28, 29] and developed further in [30–32], to confirm existing results in the literature and obtain some new large-$n$ ones. The analytic bootstrap gives the same type of results as diagrammatic methods, but simplifies the computation of certain quantities, such as scaling dimensions of spinning operators, OPE coefficients and central charges.

## 1.1 Analytic bootstrap

The analytic bootstrap relies on the fact that conformal four-point correlators can be computed from their double-discontinuities, up to potential contributions from operators with spin zero or one. The double-discontinuity measures the singularities that arise in the lightcone limit, corresponding to operators approaching pairwise null separation in Lorentzian signature, and is sensitive to operators of large spin in the operator product expansion (OPE). Spinning conformal primary operators group into twist families, where the scaling dimensions and OPE coefficients, collectively the *CFT-data*, are given by functions analytic in spin extracted from the double-discontinuity. All operators in a twist family have approximately equal value of the twist, defined as the difference between scaling dimension and spin: $\tau_\ell = \Delta_\ell - \ell$. In [33, 34] it was shown that such twist families must exist in any CFT in dimension $d > 2$, and that the CFT-data is sourced by operators appearing in the OPE decomposition of the crossed channel. Specifically, the identity operator $\mathbb{1}$ in the $\phi$ four-point function gives rise to the leading order OPE coefficients for double-twist operators $\phi \partial^\ell \phi$ with $\tau_\ell \to 2\Delta_\phi$. Other crossed-channel operators induce corrections to the CFT-data in the twist families.

Large spin perturbation theory [28, 29] combined with the Lorentzian inversion formula [35] constitutes a systematic framework for analytic bootstrap for theories with a small expansion parameter. It applies to both week coupling and strong coupling expansions, as well as to expansions in inverse number of degrees of freedom. At each order in the expansion, the whole double-discontinuity can be generated from an ansatz of contributions from a small set of crossed-channel operators, and the undetermined constants of this ansatz can later be fixed by consistency conditions, for instance conservation of symmetry currents. The method applies to a wide range of theories, and in particular it has been used to study the $\varepsilon$ expansion for the Wilson–Fisher fixed point [30] and in the large-$N$ expansion for the $O(N)$ model [32].[2] In this paper we show how to generalize these implementations to critical $\phi^4$ theories with any global symmetry group.

Consider a field $\phi$ transforming in some (vector) representation $V$ of the global symmetry. For the case of $O(m) \times O(n)$ we will take $\phi = \phi_{ar}$ in the bifundamental representation, where $a = 1, \ldots, m$ and $r = 1, \ldots, n$. Operators and twist families in the OPE decomposition of the $\phi$ four-point function will transform in all irreducible representations $R$ in the tensor product $V \otimes V$. Looking first at the $\varepsilon$ expansion, the identity $\mathbb{1}$ and bilinear operators $\phi_R^2$ will source the complete CFT-data of double-twist families in all representations to order $\varepsilon^3$. Moreover, these bilinear operators are the spin zero operators of the same twist families. Despite the fact that spin zero is beyond the formal validity of the inversion formula, it was shown in [30] that it is possible to analytically continue to the formula for the scaling dimensions in the twist families to include the scalar operators. Encouraged by this observation, we conjecture that the same is true for $\phi^4$ theories with any global symmetry, which leads to a set of quadratic equations

---

[1]Note that the $\varepsilon$ expansion can be performed with $m, n$ generic, but the resulting expressions at higher loops get very large. Below we will present $\varepsilon$ expansion results expanded in $n$.

[2]The critical $O(N)$ model has also been studied from the bootstrap perspective by the methods of multiplet recombination [36,37] and Mellin space bootstrap [38]. We briefly revisit the Mellin space bootstrap in Appendix D.

for $\Delta_{\phi_R^2}$. By solving these equations we can determine all perturbative fixed points in the $\varepsilon$ expansion consistent with the given global symmetry.[3]

For the $\phi^4$ theories with $O(N)$ symmetry, it is well-known that the large-$N$ limit admits a description in terms of a Hubbard–Stratonovich transform. In this description, the bilinear singlet operator $\phi_S^2$ gets replaced by an auxiliary field $\sigma$ of approximate dimension 2. At criticality, the large-$N$ expansion and the $\varepsilon$ expansion are compatible—using the perturbative results one can for instance confirm that $\Delta_{\phi_S^2} \to 2 + O(1/N)$. For certain global symmetry groups, the Hubbard–Stratonovich transformation can be generalized, where $N$ corresponds to a specific group parameter.[4] More precisely, if certain bilinear operators $\phi_R^2$ have dimensions approaching 2, they should be promoted to auxiliary fields $\mathcal{R}$. In large spin perturbation theory, these auxiliary fields will, together with $\mathbb{1}$, source the generalized $1/N$ expansion, and play the role same role as $\sigma$ in the treatment of the $O(N)$ model in [32].

For the symmetry group of this paper, $O(m) \times O(n)$, we will expand in $1/n$ for fixed $m$. From the results in the $\varepsilon$ expansion, we note that two representations furnish scalars that can be promoted to auxiliary fields: $\mathcal{S}$ in the singlet representation, and $\mathcal{W}_{ab}$ in the irrep that is a traceless symmetric tensor in $O(m)$ and singlet in $O(n)$. Like in the $\varepsilon$ expansion, we can derive quadratic equations, and the solutions will determine all CFT-data at order $1/n$. Apart from the free theory, we find that these equations generate all three non-trivial fixed points from the literature: the $O(mn)$ symmetric fixed point, where we have only $\mathcal{S}$; the chiral fixed point, where we have both $\mathcal{S}$ and $\mathcal{W}$; and the antichiral fixed point, where we have only $\mathcal{W}$. Amongst the results that we derive are the order $1/n$ scaling dimensions of the leading scalar operators in all (even) $O(m) \times O(n)$ irreps, and the order $1/n$ corrections to the central charges.

## 1.2 Numerical bootstrap

The numerical implementation of our work involves the study of both single and mixed correlator bootstrap systems. In our study we exclude values of scaling dimensions of various operators that are not compatible with the combined requirements of unitarity and crossing symmetry; the latter is also known as associativity of the OPE. First, we probe the correlator $\langle \phi_{ar} \phi_{bs} \phi_{ct} \phi_{du} \rangle$ for self consistency; this is our single correlator system. Previous studies of $O(2) \times O(n)$ single correlator systems have appeared in [42] and [43]. We extend their results and match with analytic predictions from the large-$n$ expansion.

For $m = 2$ and sufficiently large $n$ (e.g. $n \gtrsim 10$), we find excellent agreement between the numerical bootstrap predictions and the analytic ones. This can be clearly seen in Figs. 2 and 3. The comparison is performed by comparing the position of the kinks in the exclusion bounds in Figs. 2 and 3 with the values of the analytically predicted scaling dimensions of the corresponding operators. This reinforces the empirical notion that kinks in bootstrap bounds correspond to the position in parameter space of actual CFTs. We find that the antichiral fixed points appear as kinks in our $W$ sector, which is a representation furnished by operators that transform in the two-index traceless symmetric representation of $O(2)$ and the singlet representation of $O(n)$. The chiral fixed points coincide with kinks in our $X$ sector; operators in this representation transform as singlets of $O(2)$ and two-index traceless symmetric tensors of $O(n)$. As expected, for smaller values of $n$ the agreement between large-$n$ and numerical predictions becomes progressively worse. For $n = 2$ we find a pronounced kink that appears to correspond to a known fixed point of the $\varepsilon$ expansion as discussed in [39]. In the $n = 3$ case, there exist mild kinks that we study extensively with a mixed correlator system.

---

[3]Apart from $O(m) \times O(n)$ symmetry considered here, we have checked this reproduces all known fixed points also in the case of $MN$ [39], hypercubic and hypertetrahedral symmetry [40].

[4]For $O(N)$ symmetry, the analytic continuation in $N$ was recently put on a more solid basis using Deligne categories [41].

A mixed-correlator bootstrap for $O(2) \times O(n)$ CFTs is studied for the first time in this work.[5] It consists of probing self consistency for four-point functions involving both $\phi$ and $S$, where $S$ denotes the smallest dimension scalar operator (above $\mathbb{1}$) in the singlet representation. Our goal is to obtain closed isolated regions (islands) in parameter space, which may correspond to physical CFTs. This method has so far produced extremely accurate calculations of critical exponents in the Ising and $O(2)$ critical theories [45, 46]. Islands have also been discovered in other theories with relevance to three dimensional statistical field theory [47–50]—for a more comprehensive list of references we refer the reader to [11].

We find two sets of islands, which we identify with two qualitatively different types of fixed points. The first set corresponds to the theories predicted by the large-$n$ and $\varepsilon$ expansions; these are found by saturating bounds in the $W$ and $X$ sectors as discussed in the previous paragraph. We have found these islands for $n$ as low as 6, which appears to be in agreement with the predictions of [27]—below $n = 6$ these fixed points are expected to be nonunitary. The second set of islands appears to correspond to fixed points that have been claimed to arise after resummations of perturbative beta functions [51–53]. These are not the same fixed points that are found in the standard large $n$ and $\varepsilon$ expansions, and their existence is not widely accepted [15–17]. In our bootstrap studies these islands are found by saturating bounds in the $W$ and $Z$ sectors, where operators in the $Z$ sector transform in the antisymmetric representation of both the $O(2)$ and the $O(n)$. We find such islands for $n = 3$, which is an experimentally relevant value of $n$. The corresponding fixed points are called chiral and collinear.

For the $O(2) \times O(3)$ chiral fixed point, we find an island by saturating a bound in the $W$ sector. The associated critical exponents are

$$\beta = 0.344(5), \qquad \nu = 0.639(7). \tag{1}$$

This result is of particular interest, since the experimentally observed critical point of certain frustrated Heisenberg antiferromagnets is conjectured to be the $O(2) \times O(3)$ chiral fixed point— see [18, 51]. Our exponent $\nu$ in (1) agrees very well with experimental determinations [14, Table 37], while $\beta$ does not. Despite our results, which appear to support the existence of the $O(2) \times O(3)$ chiral and collinear fixed points, we believe that further research is required to conclusively settle outstanding issues related to criticisms of some authors in the functional RG community [15–17]. Another issue that remains unresolved is related to the assertion of some authors that the $O(2) \times O(3)$ chiral and collinear fixed points are of the focus type and thus nonunitary [53–56].

We note that in the $O(2) \times O(3)$ case, the two islands we find are consistent with a second scalar singlet that has a scaling dimension above three, i.e. the corresponding fixed points are both stable in the context of the RG. This is something that cannot hold for fixed points found in the $\varepsilon$ expansion [26, 57, 58]. We also note that all previously mentioned islands are obtained by making assumptions on the second $B$ sector operator, which contains odd-spin operators among which the first spin-one operator is the conserved vector of the $O(n)$ contained in $O(2) \times O(n)$. The fact that the allowed region presents a sensitivity to assumptions specifically on the $B$ sector was observed empirically. The dependence of bootstrap bounds on assumptions in sectors that contain conserved operators have been studied in other cases in e.g. [59–62].

The structure of this paper is as follows. In section 2 we review known perturbative results obtained in the $\varepsilon$ and large $n$ expansions. In section 3 we lay out the general formalism of the analytical bootstrap, applicable to any $\phi^4$ theory. In section 4 we apply the formalism of section 3 specifically to $O(m) \times O(n)$ theories and present explicitly numerous results. In section 5 we study $O(2) \times O(n)$ theories with the numerical bootstrap and compare to all previous results. We conclude in section 6.

---

[5]The manuscript [44], which studies the $O(3) \times O(15)$ case with the same mixed-correlator bootstrap as us, appeared on the arXiv the same day as our original submission.

## 2 Review of perturbative results

### 2.1 $\varepsilon$ expansion

Here we review results regarding CFTs with global symmetry $O_{m,n} = O(m) \times O(n)$. In the $\varepsilon = 4-d$ expansion, such CFTs are reached as endpoints of the RG flow of the two-coupling Lagrangian [13, 14, 18, 22–25]

$$\mathcal{L} = \tfrac{1}{2} \partial_\mu \phi_{ar} \partial^\mu \phi_{ar} + \tfrac{1}{8} \lambda (\phi_{ar} \phi_{ar})^2 + \tfrac{1}{24} g \, \phi_{ar} \phi_{br} \phi_{as} \phi_{bs} \,. \tag{2}$$

The $mn$ scalar fields are arranged into a matrix, $\phi_{ar}$, with row indices running from 1 to $m$ and column indices from 1 to $n$. The standard summation convention for repeated indices is used in (2). The $O(m)$ part of the symmetry group acts on the row indices and the $O(n)$ part on the column indices. Since both $O(m)$ and $O(n)$ contain the same $\mathbb{Z}_2$ symmetry generated by $\phi \to -\phi$, the correct global symmetry group is obtained by modding $O(m) \times O(n)$ out by a $\mathbb{Z}_2$. In this work we will use $O_{m,n}$, $O(m) \times O(n)$ and $O(m) \times O(n)/\mathbb{Z}_2$ interchangeably.

An equivalent Lagrangian, introduced in [24] and commonly used in subsequent literature, takes the form

$$\mathcal{L} = \tfrac{1}{2} \sum_a \partial_\mu \vec{\phi}_a \cdot \partial^\mu \vec{\phi}_a + \tfrac{1}{24} u \Big( \sum_a \vec{\phi}_a^2 \Big)^2 + \tfrac{1}{24} v \sum_{a,b} \big( (\vec{\phi}_a \cdot \vec{\phi}_b)^2 - \vec{\phi}_a^2 \vec{\phi}_b^2 \big), \tag{3}$$

where $\vec{\phi}_a$ are $m$ vectors of size $n$ each. The couplings $u, v$ of (3) are related to the couplings $\lambda, g$ of (2) by

$$u = 3\lambda + g \,, \qquad v = g \,. \tag{4}$$

Fixed points with $v < 0$ are called collinear or sinusoidal and fixed points with $v > 0$ chiral, helical or noncollinear. With $m \leqslant n$ stability of the scalar potential requires $u > 0$ if $v \leqslant 0$ and $u > (1-1/m)v$ if $v > 0$.

The number of fixed points of the Lagrangian (3) depends on the values of $m$ and $n$. There are four regimes:

(I) For $n > n^+(m)$ there are four fixed points (Gaussian, $O(mn)$, chiral, antichiral). Stable[6] fixed point: chiral.

(II) For $n^-(m) < n < n^+(m)$ there are two fixed points (Gaussian and $O(mn)$). They are both unstable.

(III) For $n_H(m) < n < n^-(m)$ there are four fixed points (Gaussian, $O(mn)$, sinusoidal, antisinusoidal). Stable fixed point: sinusoidal.

(IV) For $n < n_H(m)$ there are four fixed points (Gaussian, $O(mn)$, chiral, sinusoidal). Stable fixed point: $O(mn)$.

The Gaussian fixed point has $u = v = 0$, while the $O(mn)$ fixed point has $u > 0, v = 0$. The fully-interacting fixed points (i.e. the ones besides Gaussian and $O(mn)$) both have $uv \neq 0$ and $O_{m,n}$ global symmetry. These fixed points move around in the $\lambda$-$g$ coupling plane as $m, n$ change. For every $m$ there is a value of $n$, indicated by $n^+(m)$ above, for which the chiral and antichiral fixed points collide in the real $u$-$v$ plane and subsequently move to the complex $u$-$v$ plane. For $n > n^+(m)$ the chiral fixed point is stable, but for $n < n^+(m)$ there is no stable fixed point. However, for some $n^-(m) < n^+(m)$ two fixed points reappear in the $u$-$v$ plane—this time they are called sinusoidal and antisinusoidal because they have $v < 0$, and the sinusoidal

---

[6]A fixed point with only one relevant scalar singlet operator, namely the mass operator $\phi_{ar}\phi_{ar}$, is called stable.

fixed point is stable. Furthermore, there is a value $n^H(m) < n^-(m)$ below which the $O(mn)$ fixed point is stable, for one of the fully interacting fixed points of the $n^H(m) < n < n^-(m)$ regime crosses the $v = 0$ line and acquires $v > 0$ (chiral), while the other remains with $v < 0$ (sinusoidal). The situation is summarized in Fig. 1.

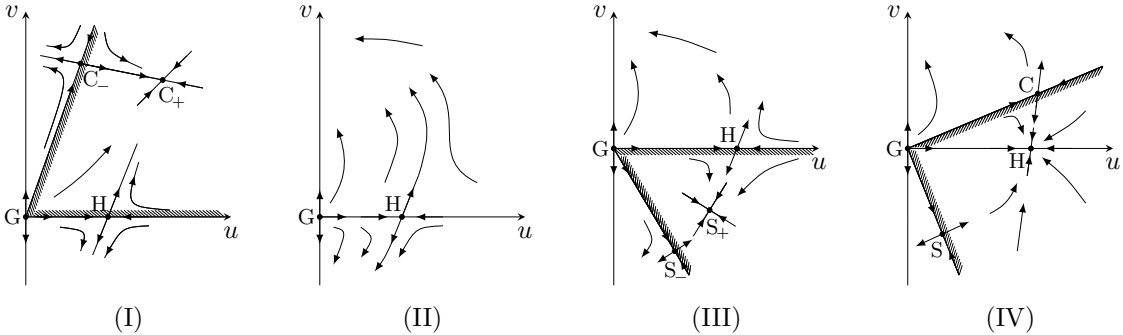

Figure 1: Flow diagrams corresponding to the various regimes mentioned in the text. The hatched regions represent the basins of attraction of the stable fixed point. This figure is a reproduction of [13, Fig. 7].

The values of $n^\pm(m)$ and $n^H(m)$ can be estimated in the $\varepsilon$ expansion [18, 19, 24, 25]:

$$n^\pm(m) = 5m + 2 \pm 2\sqrt{6(m-1)(m+2)} - \left(5m + 2 \pm \frac{25m^2 + 22m - 32}{2\sqrt{6(m-1)(m+2)}}\right)\varepsilon + O(\varepsilon^2), \quad (5)$$

$$n_H(m) = \frac{2}{m}\left(2 - \varepsilon + O(\varepsilon^2)\right). \quad (6)$$

In a recent paper, these results have been extended to six loops, or order $\varepsilon^5$ [27]. After resummation techniques are employed, the authors of [27] give, for $m = 2$, the estimates

$$n^+(2) = 5.96(19), \quad n^-(2) = 1.970(3), \quad n^H(2) = 1.462(13). \quad (7)$$

Numerical estimates for other values of $m$ can be found in [27, Table 8]. In this work we will attempt to use our bootstrap bounds to independently estimate these quantities, particularly $n^+(2)$, and compare with (7). For $m = 3$ a similar study was performed in [42], while the existence of $O(2) \times O(n)$ theories in $d = 5$ was examined with perturbative methods in [63].

As mentioned in the introduction, it has been suggested that fixed points beyond the ones we just reviewed exist in $O(2) \times O(2)$ and $O(2) \times O(3)$ theories. Confusingly, the terminology "chiral", "collinear", etc., is still used for those fixed points, depending on their sign of the coupling $v$.

For scalar theories in the $\varepsilon$ expansion below $d = 4$, it is a theorem that a stable fixed point, if it exists, is unique [26, 57, 58]. For the numerical studies in this work, we will fix $m$ to a small value, specifically $m = 2$, and obtain bounds for increasing $n$. Thus, we expect that if kinks appear at large $n$, they will be due to fixed points of regime (I). In that case, we expect from the $\varepsilon$ expansion that since the chiral fixed point is stable, the antichiral is unstable. This prediction is also expected to hold in the large-$n$ limit in $d = 3$, to which we now turn.

## 2.2 Large $n$

As mentioned in the introduction, it was realized a long time ago that $O_{m,n}$ theories admit a large-$n$ expansion [18]. For the chiral fixed point in $d = 3$, large-$n$ computations give [19–21][7]

$$\Delta_{\phi+} = \frac{1}{2} + \frac{2(m+1)}{3\pi^2}\frac{1}{n} + \frac{8(m^2-7m-26)}{27\pi^4}\frac{1}{n^2} + O\Big(\frac{1}{n^3}\Big),$$

$$\Delta_{S+} = 2 - \frac{16(m+1)}{3\pi^2}\frac{1}{n} - \frac{64(7m^2+5m-20)+108(m^2+3m+4)\pi^2}{27m^2\pi^4}\frac{1}{n^2} + O\Big(\frac{1}{n^3}\Big),$$

$$\Delta_{S'+} = 4 - \frac{32(m+1)}{3\pi^2}\frac{1}{n} + O\Big(\frac{1}{n^2}\Big),$$

$$\Delta_{S''+} = 4 - \frac{8(m+4)}{3\pi^2}\frac{1}{n} + O\Big(\frac{1}{n^2}\Big),$$

$$\Delta_{W+} = 2 - \frac{4(m+4)}{3\pi^2}\frac{1}{n} + O\Big(\frac{1}{n^2}\Big),$$

$$\Delta_{Z+} = 1 + \frac{4(m-2)}{3\pi^2}\frac{1}{n} + \frac{2\big(8(m+1)(m^2-7m-26)-27(5m+11)\pi^2\big)}{27(m+1)\pi^4}\frac{1}{n^2} + O\Big(\frac{1}{n^3}\Big), \quad (8)$$

while for the antichiral fixed point in $d = 3$ at large $n$ the results are [19, 21]

$$\Delta_{\phi-} = \frac{1}{2} + \frac{2(m-1)(m+2)}{3m\pi^2}\frac{1}{n} + \frac{8(m-1)(m+2)(m^2-8m-2)}{27m^2\pi^4}\frac{1}{n^2} + O\Big(\frac{1}{n^3}\Big),$$

$$\Delta_{S-} = 1 + \frac{16(m-1)(m+2)}{3m\pi^2}\frac{1}{n}$$
$$+ \frac{4(m-1)(m+2)\big(16(7m^2-2m+40)+27(m-2)(m+4)\pi^2\big)}{27m^2\pi^4}\frac{1}{n^2} + O\Big(\frac{1}{n^3}\Big),$$

$$\Delta_{S'''-} = 4 - \frac{8(m^2+4m-8)}{3m\pi^2}\frac{1}{n} + O\Big(\frac{1}{n^2}\Big),$$

$$\Delta_{W-} = 2 - \frac{4(m^2+4m-8)}{3m\pi^2}\frac{1}{n} + O\Big(\frac{1}{n^2}\Big). \tag{9}$$

Here we denote singlet operators in the $\phi \times \phi$ OPE with the letter $S$, operators that transform as two-index traceless symmetric tensors under $O(m)$ and singlets under $O(n)$ by the letter $W$, and operators that transform as two-index antisymmetric tensors under both $O(m)$ and $O(n)$ with the letter $Z$.[8] As usual, primes denote the order in scaling dimension of these operators, i.e. $S$ is the leading singlet, $S'$ the next-to-leading singlet and so on. We have not found large-$n$ results for $\Delta_{S'-}$ in the literature, but it is widely believed that $\Delta_{S'-} < 3$, i.e. the antichiral fixed point is unstable. By explicitly constructing singlet operators we find the ones in Table 1, where we tabulate the spectrum of the lowest dimension scalar singlet operators in the three fixed points at large $n$.

The results in (8) and (9) were obtained with the use of a Hubbard–Stratonovich transformation, extending a procedure used first in the $O(N)$ models by [64, 65]. Two Hubbard–Stratonovich auxiliary fields, $\mathcal{S}$ and $\mathcal{W}_{ab}$, are introduced in this case. $\mathcal{S}$ is a singlet, while $\mathcal{W}_{ab}$ transforms as a traceless symmetric tensor under $O(m)$ and a singlet under $O(n)$. The Lagrangian is [19, 21]

$$\mathscr{L} = \frac{1}{2}\partial_\mu\vec{\phi}_a \cdot \partial^\mu\vec{\phi}_a + \frac{1}{2}\mathcal{S}\vec{\phi}_a \cdot \vec{\phi}_a + \frac{1}{2}\mathcal{W}_{ab}\vec{\phi}_a \cdot \vec{\phi}_b - \frac{3\mathcal{S}^2}{2w} - \frac{3}{2v}\mathcal{W}_{ab}\mathcal{W}_{ab}, \tag{10}$$

---

[7]These computations are done in the more general setting of arbitrary $d$ at large $n$, but here we present the $d = 3$ results only. In [20], the operator $C$ corresponds to our $Z$, and $\eta_c$ computed there is given by $\eta_c = \Delta_Z - 1$. In [21], the operator $T$ corresponds to our $W$, and $\chi_T$ computed there is given by $\chi_T = 3 - 2\Delta_\phi - \Delta_W$.

[8]Note that there are in total nine irreps containing bilinears of $\phi$, and five of them contain scalars. We will determine dimensions of these operators in the $\varepsilon$ and large-$n$ expansions in section 4 using analytic bootstrap techniques.

where repeated indices are summed over and $w = u + (1-1/m)v$ with $u, v$ the couplings in (3). The equations of motion for $\mathcal{S}$ and $\mathcal{W}_{ab}$ can be used to go back to (3). Below we will reproduce (8) and (9) and obtain more results at order $1/n$ following the analytic bootstrap logic of [32].

Table 1: The lowest scalar singlet operators at the three non-trivial fixed points and their scaling dimensions. In the chiral fixed point, the operators $S'$ and $S''$ arise from resolving a mixing, where $\mathcal{O}$ is either $\mathcal{S}$ or $\mathcal{W}$.

| $O(mn)$ | | Chiral | | Antichiral | |
|---|---|---|---|---|---|
| $S = \sigma$ | $2 + O(\frac{1}{mn})$ | $S = \mathcal{S}$ | $2 + O(\frac{1}{n})$ | $S = \phi_S^2$ | $2\Delta_\phi + O(\frac{1}{n})$ |
| $S' = \sigma^2$ | $4 + O(\frac{1}{mn})$ | $S' = \langle [\mathcal{O}, \mathcal{O}]_{S,0,0} \rangle_1$ | $4 + O(\frac{1}{n})$ | $S' = \phi_S^4$ | $4\Delta_\phi + O(\frac{1}{n})$ |
| $S'' = [\sigma, \sigma]_{1,0}$ | $6 + O(\frac{1}{mn})$ | $S'' = \langle [\mathcal{O}, \mathcal{O}]_{S,0,0} \rangle_2$ | $4 + O(\frac{1}{n})$ | $S'' = \phi_S^6$ | $6\Delta_\phi + O(\frac{1}{n})$ |
| $S''' = \sigma^3$ | $6 + O(\frac{1}{mn})$ | | | $S''' = [\mathcal{W}, \mathcal{W}]_{S,0,0}$ | $4 + O(\frac{1}{n})$ |

# 3 Analytic bootstrap for any global symmetry

In this section we outline an implementation of the analytic bootstrap that can be applied to $\phi^4$ theories with any global symmetry. We begin with a brief review of large spin perturbation theory. This is followed by a summary of relevant results from the literature, in terms of a toolbox containing the explicit solution to the inversion problems we will encounter. We then give a general recipe for applying these tools, first to the $\varepsilon$ expansion and then to the large-$N$ expansion, where the latter is applicable to $\phi^4$ theories which admit a Hubbard–Stratonovich description.

For the analytic bootstrap, we will consider the four-point function of $\phi^i \in V$, written in the form

$$\langle \phi^i(x_1) \phi^j(x_2) \phi^k(x_3) \phi^l(x_4) \rangle = \frac{1}{(x_{12}^2 x_{34}^2)^{\Delta_\phi}} \sum_R \mathbf{T}_R^{ijkl} \mathcal{G}_R(u,v). \tag{11}$$

In this expression, $\mathbf{T}_R^{ijkl}$ are tensor structures for the irreducible representations $R \in V \otimes V$,[9] and $u, v$ are the usual cross-ratios defined by

$$u = z\bar{z} = \frac{x_{12}^2 x_{34}^2}{x_{13}^2 x_{24}^2}, \qquad v = (1-z)(1-\bar{z}) = \frac{x_{14}^2 x_{23}^2}{x_{13}^2 x_{24}^2}. \tag{12}$$

The crossing equation follows from exchanging operators at $x_2$ and $x_4$, and can we written as

$$\mathcal{G}_R(u,v) = M_{R\widetilde{R}} \left( \frac{u}{v} \right)^{\Delta_\phi} \mathcal{G}_{\widetilde{R}}(v,u), \tag{13}$$

where the explicit form of the matrix $M$ can be worked out from the tensor structures for the symmetry group under consideration. Here we choose normalizations in agreement with [32], so that the matrix in the $O(N)$ case takes the form

$$M^{O(N)} = \begin{pmatrix} 1/N & (N+2)(N-1)/(2N^2) & (1-N)/(2N) \\ 1 & (N-2)/(2N) & 1/2 \\ -1 & (N+2)/(2N) & 1/2 \end{pmatrix}, \tag{14}$$

where the representations are the singlet $S$, rank-two traceless symmetric $T$ and antisymmetric $A$ representations of $O(N)$. For any symmetry group, we reserve the letter $S$ for the singlet representation.

---

[9]More precisely, the $\mathbf{T}_R^{ijkl}$ are the projectors that can be used to decompose the four-point function into invariant subspaces labeled by $R$.

Each of the functions $\mathcal{G}_R(u,v)$ admits a decomposition in conformal blocks,

$$\mathcal{G}_R(u,v) = \sum_{\mathcal{O} \in R} c_{\phi\phi\mathcal{O}}^2 \, G_{\Delta_\mathcal{O},\ell_\mathcal{O}}(u,v), \tag{15}$$

where the sum runs over conformal primary operators $\mathcal{O}$ appearing with OPE coefficient $c_{\phi\phi\mathcal{O}}$ in the OPE $\phi^i \times \phi^j|_R$, and the conformal blocks $G_{\Delta,\ell}(u,v)$ are functions which sum up the contributions of descendants to that primary.

The OPE expansion (15) is regular in the limit $z, \bar{z} \to 0$. However, we will expand in the *lightcone limit*, defined by $z \ll 1 - \bar{z} \ll 1$. Taking $z \to 0$, the conformal blocks as functions of $z, \bar{z}$ simplify as

$$G_{\Delta,\ell}(z,\bar{z}) = z^{\frac{\Delta-\ell}{2}} k_{\frac{\Delta-\ell}{2}}(\bar{z}) + O\left(z^{\frac{\Delta+\ell}{2}+1}\right), \tag{16}$$

where $k_\beta(\bar{z}) = \bar{z}^\beta {}_2F_1(\beta,\beta;2\beta;\bar{z})$, and ${}_2F_1$ is Gauss's hypergeometric function. The lightcone limit therefore emphasizes the contribution from the operators of smallest value of the twist, defined as $\tau = \Delta - \ell$, which shows that it is useful to organize the OPE in terms of *twist families* of operators of approximately equal twist. In addition, the specific form of the hypergeometric function contains a single logarithmic divergence at $\bar{z} \to 1$, but no power or $\log^2$ divergence.

A generic CFT contains families of double-twist operators, written as $[\mathcal{O}_1, \mathcal{O}_2]_{R,n,\ell}$, where $n = 0, 1, 2, \ldots$, where $\mathcal{O}_1 \in R_1$ and $\mathcal{O}_2 \in R_2$ are operators in the theory and $R \in R_1 \otimes R_2$. For Lagrangian theories these operators have the schematic form $\mathcal{O}_1 \partial^{\mu_1} \cdots \partial^{\mu_\ell} \Box^n \mathcal{O}_2$, up to contributions from descendants. In the theories we consider, $\phi$ is near the unitarity bound, $\Delta_\phi = \frac{d-2}{2} + \gamma_\phi$, which means that the leading double-twist operators are weakly broken currents $\mathcal{J}_{R,\ell} = [\phi,\phi]_{R,0,\ell}$. Our main objective is to determine the CFT-data of these operators, which consist of their scaling dimensions $\Delta_{R,\ell}$ and their OPE coefficients $a_{R,\ell} = c_{\phi\phi\mathcal{J}_{R,\ell}}^2$. Of particular interest are the conserved currents: the stress-energy tensor $T^{\mu\nu} = \mathcal{J}_{S,2}$, and in the case of continuous global symmetry, Noether currents $J_R^\mu = \mathcal{J}_{R,1}$ in one or several representations $R$. They have conserved dimensions, $\Delta_{S,2} = d$, $\Delta_{R,1} = d - 1$ and their OPE coefficients are related to central charges $C_T$ and $C_{J_R}$ by the relations[10]

$$a_{S,2} = \frac{d^2 \Delta_\phi^2}{4(d-1)^2 C_T}, \qquad a_{R,1} = -\frac{1}{C_{J_R}}, \tag{17}$$

following from conformal Ward identities [7,66].

Our main tool is the Lorentzian inversion formula, derived in [35]:

$$C_R(\ell,\Delta) = \left(1 \pm (-1)^\ell\right) \frac{\kappa_{\Delta+\ell}}{4} \int_{[0,1]^2} dz \, d\bar{z} \, \mu(z,\bar{z}) G_{d-1+\ell,1-d+\Delta}(z,\bar{z}) \, \mathrm{dDisc}[\mathcal{G}_R(z,\bar{z})], \tag{18}$$

where the double-discontinuity dDisc is defined as the difference between the correlator and its analytic continuations around $\bar{z} = 1$, $\mathrm{dDisc}[f(\bar{z})] = f(\bar{z}) - \frac{1}{2}f^\circlearrowleft(\bar{z}) - \frac{1}{2}f^\circlearrowright(\bar{z})$. In particular, a single conformal block has vanishing double-discontinuity. The sign in (18) depends on the symmetry of $\mathcal{G}_R(z,\bar{z})$ under exchanging $x_1$ and $x_2$, and the normalization constants are given by $\kappa_\beta = \frac{\Gamma(\beta/2)^4}{2\pi^2 \Gamma(\beta)\Gamma(\beta-1)}$ and $\mu(z,\bar{z}) = |z-\bar{z}|^{d-2}(z\bar{z})^{-d}$. The poles in $\Delta$ of the function $C_R(\ell,\Delta)$ are located at the scaling dimensions of the operators $\mathcal{O}_R$, with OPE coefficients given by the residue; more precisely

$$a_{R,\ell} = -\int d\ell \oint \frac{d\Delta}{2\pi i} C_R(\ell',\Delta)\delta(\ell-\ell'). \tag{19}$$

---

[10]We use the conventions of [32], where the normalization of the conformal blocks differ with a factor $(-2)^\ell$ from e.g. [7].

The inversion formula is valid for $\ell > 1$. If a power $z^{\tau/2}$ appears in dDisc$[\mathcal{G}(z,\bar{z})]$, it signals the existence of a family of operators of twist near that value. This follows from the scaling $\mu(z,\bar{z})G_{d-1+\ell,1-d+\Delta}(z,\bar{z}) \sim z^{\frac{\ell-\Delta-2}{2}}$, which induces poles in $C_R(\ell,\Delta)$ from the $z \to 0$ limit of the $z$ integral:

$$C_R(\ell,\Delta) \sim -\frac{a_{R,\ell}}{\Delta-(\tau+\ell)}\,. \tag{20}$$

We now focus on the leading twist family in each representation, and assume that $\tau_{R,\ell} = 2\Delta_\phi + \gamma_{R,\ell}$ for small anomalous dimensions $\gamma_{R,\ell}$. In that case, following the manipulations of [35, Sec. 4], the $z$ integral can be computed and (18) reduces to the one-dimensional inversion problem given in [67]: the CFT-data $a_{R,\ell}$ and $\gamma_{R,\ell}$ are given by

$$\hat{a}_{R,\bar{h}}(\gamma_{R,\ell})^p = U^{(p)}_{R,\bar{h}} + \frac{1}{2}\partial_{\bar{h}}U^{(p+1)}_{R,\bar{h}} + \frac{1}{8}\partial^2_{\bar{h}}U^{(p+2)}_{R,\bar{h}} + \dots, \quad a_{R,\ell} = \frac{\Gamma\left(\frac{\Delta_{R,\ell}+\ell}{2}\right)^2}{\Gamma(\Delta_{R,\ell}+\ell)}\hat{a}_{R,\bar{h}}, \quad \bar{h} = \Delta_\phi + \ell\,, \tag{21}$$

where

$$U^{(p)}_{R,\bar{h}} = \frac{2^p p!\,\Gamma(\bar{h})}{\pi^2 \Gamma(2\bar{h}-1)} \int\limits_0^1 \frac{d\bar{z}}{\bar{z}^2} k_{\bar{h}}(\bar{z})\,\text{dDisc}\left[\mathcal{G}_R(z,\bar{z})\big|_{z^{\Delta_\phi}\log^p z}\right]. \tag{22}$$

These expressions were derived in [67], by assuming that $\gamma_{R,\ell} \sim g \ll 1$ and expanding all quantities in $g$. This expansion generates terms proportional to $z^{\Delta_\phi}\log^p z$ in the double-discontinuity, which under the $z$ integral are converted to higher order poles in $\Delta$ in (20). These poles are responsible for the derivatives $\partial_{\bar{h}}$ appearing in (21), following from changing variables from $(\Delta,\ell)$ to $(\tau,\bar{h})$ in (19). An alternative heuristic derivation of (22) starting from the collinear conformal blocks is given in [30].

The success of large spin perturbation theory stems from the fact that dDisc$[\mathcal{G}_R(z,\bar{z})]$ can be computed through the crossing equation (13). At each order in the expansion parameter, the whole double-discontinuity is computed from the conformal blocks of a small set of crossed-channel operators. In particular, the double-twist operators $[\phi,\phi]_{R,n,\ell}$ themselves do not contribute at leading order. This is because the power $(1-\bar{z})^{-\Delta_\phi}$ from crossing is cancelled by $(1-\bar{z})^{\frac{\tau_{n,\ell}}{2}}$ from the conformal blocks, as seen from the expansion (16) evaluated in the crossed channel, $z \to 1-\bar{z}$. Expanding $\tau_{R,n,\ell} = 2\Delta_\phi + n + \gamma_{R,n,\ell}$, we note that the first non-zero double-discontinuity arises from the term $\frac{\gamma^2_{R,n,\ell}}{8}\log^2(1-\bar{z})$. In the two cases relevant in this paper, the leading double-discontinuities will be generated by scalar operators, and the weakly broken currents $\mathcal{J}_{R,\ell} = [\phi,\phi]_{R,0,\ell}$ will only contribute at subleading order.

## 3.1 Inversion toolbox

In this section we collect all inversion formulas from the literature that are required to find the leading order CFT-data in the $\varepsilon$ expansion and in the large-$N$ expansion. The entries take the form

$$\{\mathcal{O}\} \longrightarrow U^{(0)}_{\bar{h}} + \frac{1}{2}U^{(1)}_{\bar{h}}\log z + \dots, \tag{23}$$

where $\{\mathcal{O}\}$ is a crossed channel operator or a family of such. The results have been computed from the inversion integral (22) where $\mathcal{G}_R(z,\bar{z})$ is replaced by the crossing factor $\left(\frac{u}{v}\right)^{\Delta_\phi}$ multiplied by a (sum of) conformal blocks in crossed variables. Up to an overall prefactor, the resulting functions $U^{(p)}_{\bar{h}}$ expand in inverse integer powers of the conformal spin

$$J^2 = \bar{h}(\bar{h}-1)\,, \tag{24}$$

a statement referred to in the literature as reciprocity and proved in the context of CFT in [68].

**Inversion 1.** The identity operator $\mathbb{1}$ contributes with

$$\{\mathbb{1}\} \longrightarrow \mathbf{A}[\Delta_\phi](\bar{h}) = \frac{2(2\bar{h}-1)\Gamma(\bar{h}+\Delta_\phi-1)}{\Gamma(\Delta_\phi)^2\Gamma(\bar{h}-\Delta_\phi+1)}, \tag{25}$$

which contributes to $U_{\bar{h}}^{(0)}$ only. This holds for generic $\Delta_\phi$, and applies in both the $\varepsilon$ expansion and in the large-$N$ expansion. This result can be directly computed from the integral (22) using an integral representation for $k_{\bar{h}}(\bar{z})$ (see e.g. eq. (4.7) of [35]).

**Inversion 2.** In the $\varepsilon$ expansion, a bilinear scalar $\Delta_{\phi^2} = 2\Delta_\phi + \gamma$ with OPE coefficient $c_{\phi\phi\phi^2}^2$, assuming $\gamma = g\varepsilon(1 + g^{(2)}\varepsilon + \dots)$, expanded to order $\varepsilon^3$, contributes with

$$\{\phi^2\} \longrightarrow \frac{c_{\phi\phi\phi^2}^2}{2}\gamma^2\frac{2\bar{h}-1}{J^2}\left(-1-\gamma+\varepsilon+\gamma S_1(\bar{h}-1)\right)\log z$$

$$+ \frac{c_{\phi\phi\phi^2}^2}{2}\gamma^2\frac{2\bar{h}-1}{J^4}\left(-1+(J^2\zeta_2+1)\varepsilon+(S_1(\bar{h}-1)-J^2\zeta_2-1)\gamma\right). \tag{26}$$

This was derived in eq. (2.34) of [30] using the explicit form of the scalar conformal block as an infinite sum [7] and taking the small $z$ limit. Here $S_1(\bar{h}-1)$ denotes the harmonic numbers.

**Inversion 3.** The leading contribution from a scalar $\mathcal{O}$ with $\Delta_{\mathcal{O}} = 2$, where $\Delta_\phi = \mu - 1$ in generic spacetime dimension $d = 2\mu$ is

$$\{\mathcal{O}\} \longrightarrow (\mu-2)^2 c_{\phi\phi\mathcal{O}}^2 \frac{\mathbf{A}[\mu-1](\bar{h})}{J^2}\left(-\log z + \mathbf{S_1}[\mu-1](\bar{h}) - \frac{1}{J^2}\right), \tag{27}$$

where

$$\mathbf{S_1}[\alpha](\bar{h}) = 2S_1(\bar{h}-1) - S_1(\bar{h}-2+\alpha) - S_1(\bar{h}-\alpha). \tag{28}$$

This was derived in eqs. (2.25) and (2.29) of [32].

**Inversion 4.** The leading contribution from an infinite sum over $\ell \in \mathbb{I}_\pm$ of broken currents $\mathcal{J}_\ell$ with anomalous dimensions $\gamma_\ell = \frac{\kappa}{J^2}$ and OPE coefficients $\alpha a_{0,\ell}^{\text{GFF}}|_{\mu-1}$ (where $a_{n,\ell}^{\text{GFF}}|_{\Delta_\phi}$ are the generalized free field OPE coefficients given in (84)) is

$$\{\mathcal{J}_\ell\}_{\ell\in\mathbb{I}_\pm,\gamma=\frac{\kappa}{J^2}} \longrightarrow -\alpha\kappa^2\frac{2\bar{h}-1}{2(\mu-2)^2J^2}\left[\pm 1 + (\mu-2)\pi\csc(\pi\mu)\right]\ln z + E_\pm, \quad \mathbb{I}_\pm = \begin{cases} \{0,2,4,\dots\} \\ \{1,3,5,\dots\} \end{cases}. \tag{29}$$

Here $E_\pm$ are lengthy expressions given explicitly in (85). This formula was derived in [32] by summing over blocks on the unitarity bound and subsequently inverting the sum.

## 3.2 General solution in the $\varepsilon = 4 - d$ expansion

Consider first the contribution from the identity operator, appearing in the singlet ($S$) representation. This will give rise to the leading contribution to $U_{R,\bar{h}}^{(0)}$ in all representations. Since this is the only operator contributing until order $\varepsilon^2$, we get, using Inversion 1,

$$U_{R,\bar{h}}^{(0)} = M_{RS}\mathbf{A}[\Delta_\phi](\bar{h}) + \mathrm{O}(\varepsilon^2), \tag{30}$$

where $\Delta_\phi = 1 - \varepsilon/2 + \gamma_\phi$ with $\gamma_\phi = \mathrm{O}(\varepsilon^2)$. From this expression, the leading order OPE coefficients can be extracted:

$$c_{\phi\phi\mathcal{J}_{R,\ell}}^2 = \frac{2\Gamma(\ell+1)^2}{\Gamma(2\ell+1)}M_{RS} + \mathrm{O}(\varepsilon). \tag{31}$$

Here $\ell$ takes even (odd) values for $R$ being an even (odd) representation. The scalar bilinears $\phi_R^2$ in the even representations have OPE coefficients

$$c_{\phi\phi\phi_R^2}^2 = 2M_{RS} + \mathrm{O}(\varepsilon). \tag{32}$$

These scalars are the next operators to contribute to the double-discontinuity. Assume that they have dimension $\Delta_{\phi_R^2} = 2\Delta_\phi + g_R\varepsilon + \mathrm{O}(\varepsilon^2)$. Then, using Inversion 2 we get the order $\varepsilon^2$ corrections

$$U_{R,\bar{h}}^{(1)} = -M_{RS}\,\Gamma_R^{\{2\}}\frac{2(2\bar{h}-1)}{J^2}\varepsilon^2 + \mathrm{O}(\varepsilon^3), \tag{33}$$

$$U_{R,\bar{h}}^{(0)} = M_{RS}\,\mathbf{A}[\Delta_\phi](\bar{h}) - M_{RS}\,\Gamma_R^{\{2\}}\frac{2\bar{h}-1}{J^4}\varepsilon^2 + \mathrm{O}(\varepsilon^3), \tag{34}$$

where

$$\Gamma_R^{\{2\}} = \frac{1}{M_{RS}}\sum_{\widetilde{R}\text{ even}} M_{R\widetilde{R}}\,g_{\widetilde{R}}^2\,M_{\widetilde{R}S}. \tag{35}$$

Using (21) we can thus write down the leading correction to the anomalous dimension,

$$\Delta_{R,\ell} = 2\Delta_\phi + \ell + \gamma_R(\bar{h}), \qquad \gamma_R(\bar{h}) = -\frac{\Gamma_R^{\{2\}}\varepsilon^2}{J^2}, \tag{36}$$

where $\bar{h} = \Delta_\phi + \ell$ and $J^2 = \bar{h}(\bar{h}-1)$.

Next, as observed in [30], we assume that it is possible to analytically continue the result $\gamma_R(\bar{h})$ to spin zero, by making the change of variables $\bar{h} \to \bar{h}_{\mathrm{f}} = \frac{\Delta+\ell}{2}$, i.e. we replace the *bare* with the *full* conformal spin (eigenvalue of the quadratic Casimir). For spin zero we should evaluate this at $\bar{h}_{\mathrm{f}} = \Delta_{\phi_R^2}/2 = 1 - \varepsilon/2 + g_R\varepsilon/2 + \mathrm{O}(\varepsilon^2)$. This leads to a system of equations

$$g_R \stackrel{!}{=} \gamma_R(\bar{h})\big|_{\bar{h}=\Delta_\phi+\frac{g_R}{2}}, \qquad R\text{ even}, \tag{37}$$

at order $\varepsilon$, where now one power of $\varepsilon$ in the $\gamma_R(\bar{h})$ cancels against the factor $\bar{h}_{\mathrm{f}}-1 = (g_R-1)\varepsilon/2$ in the denominator. This simplifies to

$$M_{RS}\,g_R(g_R-1) + 2\sum_{\widetilde{R}\text{ even}} M_{R\widetilde{R}}M_{\widetilde{R}S}\,g_{\widetilde{R}}^2 = 0, \qquad R\text{ even}, \tag{38}$$

which is a system of $k$ quadratic equations for the $k$ constants $g_R$, where $k$ is the number of even representations, or equivalently the number of scalar bilinears. Solving (38) gives all possible fixed points in the $\varepsilon$ expansion with the given symmetry.

As an example, consider the $O(N)$ model with crossing matrix (14). The even representations are $S$ and $T$, and the bilinear scalars are $\phi_S^2 = \phi^i\phi^i$ and $\phi_T^2 = \phi^{\{i}\phi^{j\}}$. There are two solutions to (38), $g_S = g_T = 0$, which is the Gaussian theory, and

$$g_S = \frac{N+2}{N+8}, \qquad g_T = \frac{2}{N+8}, \tag{39}$$

which is exactly the critical $O(N)$ model [31]. With these values we have $\Gamma_S^{\{2\}} = \frac{3(N+2)}{(N+8)^2}$, $\Gamma_T^{\{2\}} = \frac{N+6}{(N+8)^2}$ and $\Gamma_A^{\{2\}} = \frac{N+2}{(N+8)^2}$.

The singlet spin-two current in any global symmetry group is the stress-energy tensor with dimension $\Delta_{S,2} = d = 4 - \varepsilon$. This gives the constraint

$$\gamma_\phi^{(2)} = \tfrac{1}{12}\Gamma_S^{\{2\}}, \tag{40}$$

where $\Delta_\phi = 1 - \varepsilon/2 + \gamma_\phi^{(2)}\varepsilon^2 + \mathrm{O}(\varepsilon^3)$. Using this we write down the full dimension of the broken currents to order $\varepsilon^2$:

$$\Delta_{R,\ell} = 2 - \varepsilon + \ell + 2\gamma_\phi^{(2)}\varepsilon^2 - \frac{\Gamma_R^{\{2\}}\varepsilon^2}{\ell(\ell+1)} + \mathrm{O}(\varepsilon^3). \tag{41}$$

The OPE coefficients are extracted using (21),

$$a_{R,\ell} = M_{RS}a_{0,\ell}^{\mathrm{GFF}} + M_{RS}\frac{\Gamma_R^{\{2\}}\varepsilon^2}{\ell(\ell+1)}\left(S_1(2\ell) - S_1(\ell) + \frac{1}{\ell+1}\right)\frac{2\Gamma(\ell+1)^2}{\Gamma(2\ell+1)} + \mathrm{O}(\varepsilon^3), \tag{42}$$

where $a_{0,\ell}^{\mathrm{GFF}}$ is the generalized free field OPE coefficients for $\Delta_\phi = 1 - \varepsilon/2 + \gamma_\phi^{(2)}\varepsilon^2$ expanded to order $\varepsilon^2$, which we give in (83) in Appendix B.

From the spin two singlet OPE coefficient we can extract the central charge correction given by (17)

$$\frac{C_T}{C_{T,\mathrm{free}}} = 1 - \frac{5\gamma_\phi^{(2)}}{3}\varepsilon^2 + \mathrm{O}(\varepsilon^3) = 1 - \frac{5\Gamma_S^{\{2\}}}{36}\varepsilon^2 + \mathrm{O}(\varepsilon^3), \tag{43}$$

which is consistent with [25, Eq. (E.1)]. We emphasize that the considerations here are valid with any global symmetry group. The input needed to specialize to a given symmetry group is the crossing matrix $M_{R\widetilde{R}}$ and the division of the representations into even and odd spin. By solving the system of equations (38) one finds all fixed points in the $\varepsilon$ expansion compatible with that symmetry group and derives the leading (order $\varepsilon$) anomalous dimensions of the bilinear scalars. Conservation of the stress-energy tensor allows one to compute the leading (order $\varepsilon^2$) anomalous dimension of $\phi$.

In one or several of the odd representations $R$, the current at spin $\ell = 1$ may be conserved, being a generator of global symmetry. This therefore gives further constraints $\Delta_{R,\ell} = d - 1$, which must be explicitly checked. By (17) the corresponding OPE coefficient is related to the $C_J$ of that symmetry current:

$$\frac{C_{J_R}}{C_{J_R,\mathrm{free}}} = 1 - 3\gamma_\phi^{(2)}\varepsilon^2 + \mathrm{O}(\varepsilon^3) = 1 - \frac{3\Gamma_R^{\{2\}}}{4}\varepsilon^2 + \mathrm{O}(\varepsilon^3). \tag{44}$$

Let us discuss the extension to higher orders in the $\varepsilon$ expansion. To order $\varepsilon^3$, the operators contributing with a nonzero double-discontinuity are the same as at the previous order, namely the bilinear scalars $\phi_R^2$. At higher orders, infinite families of operators contribute. In the $O(N)$ model, the only such families at order $\varepsilon^4$ are operators of approximate twist 2 and 4, and subsequently the problem was solved there. We expect that this generalises to any global symmetry. However, to compute the contribution from approximate twist 4 requires detailed knowledge of the operator content of the theory in question. This was done in the case of the $O(N)$ model in [31]. To order $\varepsilon^5$ the same operators will contribute but now with subleading corrections. To work this out, even in the $N = 1$ (Ising) case, is still an open problem.

We have seen that at order $\varepsilon^2$ all constants that enter the problem can be fixed using continuation to spin zero and conservation of the stress-energy tensor. This is no longer true at higher orders. At order $\varepsilon^3$ a total of $2k + 1$ new constants appear: $\gamma_\phi^{(3)}$, the second order correction to $\gamma_{\phi_R^2} = g_R\varepsilon(1 + g_R^{(2)}\varepsilon) + \dots$, and the corrections $\alpha_R$ to the OPE coefficients defined by

$$c_{\phi\phi\phi_R^2}^2 = 2M_{RS}(1 + \alpha_R\varepsilon) + \mathrm{O}(\varepsilon^2). \tag{45}$$

Based on experience from the $O(N)$ model [31], the order $\varepsilon^2$ continuation to spin zero requires order $\varepsilon^4$ results for the currents, so the only new equations at order $\varepsilon^3$ are the conservation

of the symmetry currents (including the stress-energy tensor). In general this will not provide enough equations to fix all constants, but in many cases we can still make progress. Firstly, we will use that the correction to the OPE coefficient takes the form $\alpha_R = -g_R$. This holds for any global symmetry, and follows from analytic bootstrap in Mellin space, as we show in Appendix D. Secondly, the second order corrections $g_R^{(2)}$ to the bilinear scalar dimensions are in many cases known from the literature, and one can proceed using these as input.

Using Inversion 2, it is straightforward to derive expressions for $U_{R,\bar{h}}^{(p)}$ at order $\varepsilon^3$. The anomalous dimensions extracted from these expressions take the form

$$\gamma_R(\bar{h}) = -\frac{\Gamma_R^{\{2\}}}{J^2}\varepsilon^2 + \frac{\Gamma_R^{\{2\}} - 2\Gamma_R^{\{2,1\}} + \big(\Gamma_R^{\{3\}} - \Gamma_R^{\{2\}}\big)S_1(\bar{h}-1)}{J^2}\varepsilon^3 + O(\varepsilon^4), \qquad (46)$$

where $\bar{h} = 1 - \frac{\varepsilon}{2} + \ell + O(\varepsilon^2)$ and

$$\Gamma_R^{\{3\}} = \frac{1}{M_{RS}} \sum_{\widetilde{R} \text{ even}} M_{R\widetilde{R}} g_{\widetilde{R}}^3 M_{\widetilde{R}S}, \qquad \Gamma_R^{\{2,1\}} = \frac{1}{M_{RS}} \sum_{\widetilde{R} \text{ even}} M_{R\widetilde{R}} g_{\widetilde{R}}^2 g_{\widetilde{R}}^{(2)} M_{\widetilde{R}S}. \qquad (47)$$

From the corresponding expression for the OPE coefficients using Inversion 2, we can extract the central charge correction:

$$\frac{C_T}{C_{T,\text{free}}} = 1 - \frac{5}{3}\big(\gamma_\phi^{(2)}\varepsilon^2 + \gamma_\phi^{(3)}\varepsilon^3\big) - \frac{29}{18}\gamma_\phi^{(2)}\varepsilon^3 + \frac{5}{48}\Gamma_S^{\{3\}}\varepsilon^3 + O(\varepsilon^4). \qquad (48)$$

Here we used that the stress-energy tensor conservation eliminates the dependence on $g_R^{(2)}$ in favour of $\gamma_\phi^{(3)}$. For two-coupling theories as considered in [25] we may find

$$\Gamma_S^{\{3\}} = \tfrac{1}{90}(N+2)\big(58(a g_*^2 + 6\lambda_*^2) - 258(N+8)\lambda_*^3 - 129 a(b g_* + 6\lambda_*)g_*^2\big), \qquad (49)$$

where $\lambda_*, g_*$ are the coefficients of the order-$\varepsilon$ values of the two couplings at the fixed point, i.e. $\lambda = \lambda_*\varepsilon + O(\varepsilon^2)$ and $g = g_*\varepsilon + O(\varepsilon^2)$, and $a, b$ are defined in [25, Eq. (5.2)]. For the $O(N)$ model, where $\lambda_* = 1/(N+8)$ and $g_* = 0$, this gives $\Gamma_S^{\{3\}} = (N+2)/(N+8)^2$, in complete agreement with (47).

Similarly, for the current central charges we derive the expression

$$\frac{C_{J_R}}{C_{J_R,\text{free}}} = 1 - 3\big(\gamma_\phi^{(2)}\varepsilon^2 + \gamma_\phi^{(3)}\varepsilon^3\big) - \frac{9}{4}\gamma_\phi^{(2)}\varepsilon^3 + \frac{1}{4}\Gamma_R^{\{3\}}\varepsilon^3 + O(\varepsilon^4). \qquad (50)$$

## 3.3 General solution in the large $N$ expansion

Let us now describe the computation of CFT-data in the large-$N$ expansion for a generic symmetry group, parametrised by some number $N$. Compared to the $\varepsilon$ expansion the situation is a bit more complicated, since the parameter $N$ enters in the crossing matrix $M_{R\widetilde{R}}$ itself. In a given even representation $R$, we assume that there are two possibilities for the smallest dimension scalar. It is either a scalar bilinear $\phi_R^2$ with dimension $2\Delta_\phi + O(N^{-1})$, or a Hubbard–Stratonovich field $\mathcal{R}$ with dimension $2 + O(N^{-1})$. If one has access to results in the $\varepsilon$ expansion, one can assess the situation by taking the large $N$ limit of the order $\varepsilon$ scalar dimensions. For instance, in the $O(N)$ model we get, using (39),

$$\Delta_{\phi_S^2} = 2 - \varepsilon + g_S\varepsilon \to 2 + O(N^{-1}), \qquad \Delta_{\phi_T^2} = 2 - \varepsilon + g_T\varepsilon \to d - 2 + O(N^{-1}), \qquad (51)$$

so we see that the singlet representation admits a Hubbard–Stratonovich field $\mathcal{S}$ (in the literature denoted by $\sigma$), but not the traceless symmetric representation (whence we keep the notation $\phi_T^2$). We assume that the Hubbard–Stratonovich fields $\mathcal{R}$ have dimension $\Delta_\mathcal{R} = 2 + O(N^{-1})$ and OPE coefficient $c_{\phi\phi\mathcal{R}}^2 = a_\mathcal{R}/N + O(N^{-2})$.

In order to provide some structure of the subsequent computations we define the following subsets of the representations in $V \otimes V = \text{I} \cup \text{II}$:

- Group I: Representations whose only corrections at order $1/N$ come from crossed channel Hubbard–Stratonovich fields.

- Group II: Representations where the corrections at order $1/N$ come from Hubbard–Stratonovich fields as well as from broken currents in Group I representations in the crossed channel.

- Group III: Representations that admit a Hubbard–Stratonovich field. Typically III $\subset$ II.

As an example, in the $O(N)$ model we have $S \in \text{II} \cap \text{III}$ and $T, A \in \text{I}$. Our strategy will then be the following. First, as in the $\varepsilon$ expansion, the identity operator creates the leading contribution to $U_{R,\bar{h}}^{(0)}$ for all representations. Next we turn to the representations in Group I. The contributions from Hubbard–Stratonovich fields will give the order $1/N$ anomalous dimensions in these representations. Using Inversion 3 we see that these corrections will be proportional to $1/J^2$. Finally we turn to representations in the Group II. Here we get contributions from both the Hubbard–Stratonovich fields, using Inversion 3, and from the currents in Group I. Due to the particular form of the anomalous dimensions of these currents, we can use Inversion 4 to find the complete order $1/N$ CFT-data.

The expressions will depend on $|\text{III}| + 1$ free parameters: the OPE coefficients $a_{\mathcal{R}} = c_{\phi\phi\mathcal{R}}^2$ for $R \in \text{III}$, and the leading order anomalous dimension of $\phi$. The only consistency conditions available to fix these constants are the conservation of the symmetry currents (including the stress-energy tensor), and depending on the number of conserved currents this may or may not be enough. As in the order $\varepsilon^3$ results above, literature values can be used to fix the remaining constants if the conservation equations are not sufficient. Finally, the leading anomalous dimensions of the Hubbard–Stratonovich fields may be extracted by imposing a shadow relation similar to the one observed in the $O(N)$ model [32]:

$$\Delta_{\mathcal{R}} + \Delta_{R,0} \overset{!}{=} d. \tag{52}$$

Let us now execute the strategy in full generality. The contribution from the identity operator gives

$$U_{R,\bar{h}}^{(0)} = M_{RS} \mathbf{A}[\Delta_\phi](\bar{h}), \tag{53}$$

where now $\Delta_\phi = \mu - 1 + \gamma_\phi^{(1)}/N + O(N^{-2})$ with $d = 2\mu$. For the representations in Group I we get the contributions from Hubbard–Stratonovich fields in Group III. Using Inversion 3 we get

$$U_{R,\bar{h}}^{(1)} = -\sum_{\widetilde{R} \in \text{III}} M_{R\widetilde{R}} 2(\mu-2)^2 \frac{a_{\widetilde{R}}}{N} \frac{\mathbf{A}[\mu-1](\bar{h})}{J^2}, \quad R \in \text{I}, \tag{54}$$

and a corresponding expression for $U_{R,\bar{h}}^{(0)}$. From this we extract the order $N^{-1}$ anomalous dimensions of currents in Group I representations:

$$\gamma_{R,\bar{h}} = -\frac{2(\mu-2)^2 K_R}{J^2 N} + O(N^{-2}), \quad K_R = \frac{1}{M_{RS}} \sum_{\widetilde{R} \in \text{III}} M_{R\widetilde{R}} a_{\widetilde{R}}, \quad R \in \text{I}, \tag{55}$$

where the scaling dimensions are given by $\Delta_{R,\ell} = 2\Delta_\phi + \ell + \gamma_{R,\ell}$ In step 3 we consider the second group of operators, II. They get contributions both from $\mathcal{R}$ for $R \in \text{III}$ and from $\mathcal{J}_{R,\ell}$ for $R \in \text{I}$. We get

$$U_{R,\bar{h}}^{(1)} = -\sum_{\widetilde{R} \in \text{III}} 2 M_{R\widetilde{R}} (\mu-2)^2 \frac{a_{\widetilde{R}}}{N} \frac{\mathbf{A}[\mu-1](\bar{h})}{J^2}$$

$$-\sum_{\widetilde{R}_\pm \in \mathrm{I}} 4 M_{R\widetilde{R}} K_{\widetilde{R}}^2 M_{\widetilde{R}S} \frac{(\mu-2)^2(2\bar{h}-1)}{J^2 N^2}(\pm 1 + (\mu-2)\pi\csc(\pi\mu)), \quad R \in \mathrm{II}, \qquad (56)$$

where the $+$ $(-)$ sign is used if the operators in the $\widetilde{R}$ representations have even (odd) spin. This means that the anomalous dimensions of the group II double-twist operators are

$$\gamma_{R,\ell} = -\frac{2(\mu-2)^2 K_R}{J^2 N} - \frac{\widehat{K}_R}{J^2 N^2} \frac{(\mu-2)^2 \Gamma(\mu+1)^2 \Gamma(\ell+1)}{\Gamma(2\mu+\ell-3)}, \quad R \in \mathrm{II}. \qquad (57)$$

In the above expressions $J^2 = (\mu-1+\ell)(\mu-2+\ell)$ and

$$\widehat{K}_R = \frac{1}{M_{RS}} \sum_{\widetilde{R}_\pm \in \mathrm{I}} 2 M_{R\widetilde{R}} K_{\widetilde{R}}^2 M_{\widetilde{R}S}(\pm 1 + (\mu-2)\pi\csc(\pi\mu)), \quad R \in \mathrm{II}. \qquad (58)$$

As an example, let us explicitly evaluate $K_R$ and $\widehat{K}_R$ in the $O(N)$ model. We get

$$K_R = a_{\mathcal{S}}, \; R = S,T,A, \qquad \widehat{K}_S = 2N(\mu-2)\pi\csc(\pi\mu)a_{\mathcal{S}}^2 + O(N^0). \qquad (59)$$

We have two conservation equations, a linear relation due to the global symmetry current $\Delta_{A,1} = d-1$, and a quadratic relation due to the stress tensor $\Delta_{S,2} = d$. There are two solutions, free theory $a_{\mathcal{S}} = \gamma_\phi^{(1)} = 0$ and the known $O(N)$ model result [29, 32]

$$\gamma_\phi^{(1)} = \frac{(\mu-2)^2}{\mu(\mu-1)} a_{\mathcal{S}} = \eta_1^{O(N)} := \frac{(\mu-2)\Gamma(2\mu-1)}{\Gamma(\mu)\Gamma(\mu+1)\pi\csc(\pi\mu)}. \qquad (60)$$

The extension to subleading orders in $1/N$ is a complicated task, which was achieved in [32] for the $T$ and $A$ representations. This involved computing the contributions from operators $[\sigma,\sigma]_{n,\ell}$, which was found in the form of a Mellin space amplitude, using OPE coefficients derived from the mixed correlator $\langle\phi\phi\sigma\sigma\rangle$. We do not attempt to generalize it for generic global symmetry group.

## 4 Analytic bootstrap of $O(m) \times O(n)$ CFTs

In this section we will apply the methods of section 3 to the global symmetry group $O(m)\times O(n)$.

The crossing matrix $M_{R\widetilde{R}}$ in the basis $\{S, W, X, Y, Z, A, B, C, D\}$, where in terms of products of representations of each orthogonal group we have

$$S = (S,S), \quad W = (T,S), \quad X = (S,T), \quad Y = (T,T), \quad Z = (A,A), \qquad (61)$$

$$A = (A,S), \quad B = (S,A), \quad C = (A,T), \quad D = (T,A), \qquad (62)$$

where by $S = (S,S)$ we mean that we take the singlet of each of $O(m)$ and $O(n)$ to form the representation $S$ of $O_{m,n} = O(m) \times O(n)$ and similarly for the rest, takes the form

$$\begin{pmatrix}
\frac{1}{mn} & \frac{m^2+m-2}{m^2 n} & \frac{n^2+n-2}{mn^2} & \frac{(m^2+m-2)(n^2+n-2)}{m^2 n^2} & \frac{(m-1)(n-1)}{mn} & \frac{m-1}{mn} & \frac{n-1}{mn} & \frac{(m-1)(n^2+n-2)}{mn^2} & \frac{(m^2+m-2)(n-1)}{m^2 n} \\
\frac{1}{2n} & \frac{m-2}{2mn} & \frac{n^2+n-2}{2n^2} & \frac{(m-2)(n^2+n-2)}{2mn^2} & -\frac{n-1}{2n} & -\frac{1}{2n} & \frac{n-1}{2n} & -\frac{n^2+n-2}{2n^2} & \frac{(m-2)(n-1)}{2mn} \\
\frac{1}{2m} & \frac{m^2+m-2}{2m^2} & \frac{n-2}{2mn} & \frac{(m^2+m-2)(n-2)}{2m^2 n} & -\frac{m-1}{2m} & \frac{m-1}{2m} & -\frac{1}{2m} & \frac{(m-1)(n-2)}{2mn} & -\frac{m^2+m-2}{2m^2} \\
\frac{1}{4} & \frac{m-2}{4m} & \frac{n-2}{4n} & \frac{(m-2)(n-2)}{4mn} & \frac{1}{4} & -\frac{1}{4} & -\frac{1}{4} & -\frac{n-2}{4n} & -\frac{m-2}{4m} \\
\frac{1}{4} & -\frac{m+2}{4m} & -\frac{n+2}{4n} & \frac{(m+2)(n+2)}{4mn} & \frac{1}{4} & \frac{1}{4} & \frac{1}{4} & -\frac{n+2}{4n} & -\frac{m+2}{4m} \\
\frac{1}{2n} & -\frac{m+2}{2mn} & \frac{n^2+n-2}{2n^2} & -\frac{(m+2)(n^2+n-2)}{2mn^2} & \frac{n-1}{2n} & \frac{1}{2n} & \frac{n-1}{2n} & \frac{n^2+n-2}{2n^2} & -\frac{(m+2)(n-1)}{2mn} \\
\frac{1}{2m} & \frac{m^2+m-2}{2m^2} & -\frac{n+2}{2mn} & -\frac{(m^2+m-2)(n+2)}{2m^2 n} & \frac{m-1}{2m} & \frac{m-1}{2m} & \frac{1}{2m} & -\frac{(m-1)(n+2)}{2mn} & \frac{m^2+m-2}{2m^2} \\
\frac{1}{4} & -\frac{m+2}{4m} & \frac{n-2}{4n} & -\frac{(m+2)(n-2)}{4mn} & -\frac{1}{4} & \frac{1}{4} & -\frac{1}{4} & \frac{n-2}{4n} & \frac{m+2}{4m} \\
\frac{1}{4} & \frac{m-2}{4m} & -\frac{n+2}{4n} & -\frac{(m-2)(n+2)}{4mn} & -\frac{1}{4} & -\frac{1}{4} & \frac{1}{4} & \frac{n+2}{4n} & \frac{m-2}{4m}
\end{pmatrix}.$$

$$\qquad (63)$$

This matrix is simply the tensor product $M^{O(m)} \otimes M^{O(n)}$ for $M^{O(N)}$ given in (14). The representations are either even ($S$, $W$, $X$, $Y$ and $Z$) or odd ($A$, $B$, $C$ and $D$) under $x_1 \leftrightarrow x_2$. The even (odd) representations have intermediate operators of even (odd) spins.

## 4.1 Results in the $\varepsilon$ expansion

In the $\varepsilon$ expansion, the operators $\phi_R^2$ in the five even representations $R$ introduce corrections to weakly broken currents in all nine representations. Solving equations (38) for the constants $g_R$ we find four sets of solutions, corresponding to the free theory and to the $O(mn)$ (Heisenberg), chiral and antichiral fixed points. For the latter two fixed points, of interest to this paper, the explicit expressions for the $g_R$ are rather complicated, containing square roots. We give complete results in an ancillary data file, which we describe in Appendix A. For presentation purposes the expressions in the $\varepsilon$ expansion in this section are expanded for at large $n$, but at each order in $\varepsilon$ presented here the complete function of $m$ and $n$ has been determined.

The constants $g_R$ correspond to the scaling dimensions of the scalar operators, which take the form

$$
\begin{aligned}
\Delta_{\phi_S^2+} &= 2 - 3(m+1)\frac{\varepsilon}{n} - 3\left(m^2 - 3m - 14\right)\frac{\varepsilon}{n^2} + O\left(\varepsilon^2, n^{-3}\right), \\
\Delta_{\phi_W^2+} &= 2 - (m+3)\frac{\varepsilon}{n} + \left(m^2 + 7m + 42\right)\frac{\varepsilon}{n^2} + O\left(\varepsilon^2, n^{-3}\right), \\
\Delta_{\phi_X^2+} &= 2 - \varepsilon + (m+1)\frac{\varepsilon}{n} - \left(m^2 + 5m + 10\right)\frac{\varepsilon}{n^2} + O\left(\varepsilon^2, n^{-3}\right), \\
\Delta_{\phi_Y^2+} &= 2 - \varepsilon + \frac{\varepsilon}{n} - (m+10)\frac{\varepsilon}{n^2} + O\left(\varepsilon^2, n^{-3}\right), \\
\Delta_{\phi_Z^2+} &= 2 - \varepsilon - \frac{\varepsilon}{n} + (m-2)\frac{\varepsilon}{n^2} + O\left(\varepsilon^2, n^{-3}\right),
\end{aligned}
\tag{64}
$$

for the chiral fixed point, and

$$
\begin{aligned}
\Delta_{\phi_S^2-} &= 2 - \varepsilon + \frac{3(m-1)(m+2)}{m}\frac{\varepsilon}{n} + \frac{3(m-1)(m+2)\left(m^2 - 4m + 16\right)}{m^2}\frac{\varepsilon}{n^2} + O\left(\varepsilon^2, n^{-3}\right), \\
\Delta_{\phi_W^2-} &= 2 - \frac{(m-2)(m+5)}{m}\frac{\varepsilon}{n} + \frac{(m-4)\left(m^3 + 11m^2 + 14m - 40\right)}{m^2}\frac{\varepsilon}{n^2} + O\left(\varepsilon^2, n^{-3}\right), \\
\Delta_{\phi_X^2-} &= 2 - \varepsilon + \frac{(m-1)(m+2)}{m}\frac{\varepsilon}{n} - \frac{(m-1)(m+2)\left(m^2 + 4m - 16\right)}{m^2}\frac{\varepsilon}{n^2} + O\left(\varepsilon^2, n^{-3}\right), \\
\Delta_{\phi_Y^2-} &= 2 - \varepsilon + \frac{m-2}{m}\frac{\varepsilon}{n} - \frac{m^3 - 2m^2 - 24m + 32}{m^2}\frac{\varepsilon}{n^2} + O\left(\varepsilon^2, n^{-3}\right), \\
\Delta_{\phi_Z^2-} &= 2 - \varepsilon - \frac{m+2}{m}\frac{\varepsilon}{n} + \frac{(m+2)\left(m^2 + 8m - 16\right)}{m^2}\frac{\varepsilon}{n^2} + O\left(\varepsilon^2, n^{-3}\right),
\end{aligned}
\tag{65}
$$

for the antichiral fixed point. These results agree with [25, Eqs. (5.92) and (5.93)].[11] The operators $\phi_S^2$, $\phi_W^2$ and $\phi_Z^2$ correspond to $S$, $W$ and $Z$, respectively, in (8) and (9).

Having identified the fixed points we move on to a determination of the CFT-data to order $\varepsilon^3$. As described in the previous section, we need to take as input the second order corrections $g_R^{(2)}$ to the anomalous dimensions $\gamma_{\phi_R^2}$ of bilinear scalars, given in [25].

We present only a subset of the data computed at order $\varepsilon^3$. The dimension of $\phi$,

$$
\Delta_{\phi+} = 1 - \frac{\varepsilon}{2} + \frac{m+1}{8n}\varepsilon^2 - \frac{2m^2 + 9m + 17}{8n^2}\varepsilon^2 - \frac{m+1}{32n}\varepsilon^3 + \frac{14m^2 + 57m + 101}{32n^2}\varepsilon^3 + O\left(\varepsilon^4, n^{-3}\right),
$$

---

[11]In the notation of [25], $\rho_1, \rho_2, \rho_3, \rho_4$ correspond to $\phi_X^2, \phi_W^2, \phi_Y^2, \phi_Z^2$, respectively. There is a typo in $\gamma_{\rho_1\pm}$ in [25, Eq. (5.93)]: the sign before the $1/n$ term there should be positive.

$$\Delta_{\phi-} = 1 - \frac{\varepsilon}{2} + \frac{(m-1)(m+2)}{8mn}\left(1 - \frac{2m^2 + 7m - 22}{mn}\right)\varepsilon^2$$
$$- \frac{(m-1)(m+2)}{32mn}\left(1 - \frac{14m^2 + 43m - 158}{mn}\right)\varepsilon^3 + O\left(\varepsilon^4, n^{-3}\right), \quad (66)$$

agrees with the literature values [19], whereas the results of the spinning operators are new, as far as we are aware. These include the dimensions of the non-conserved spin-one operators

$$\Delta_{C,1+} = 3 - \varepsilon + \frac{m+2}{4n}\varepsilon^2 - \frac{m^2 + 6m + 8}{2n^2}\varepsilon^2 - \frac{m+2}{16n}\varepsilon^3 + \frac{7m^2 + 36m + 44}{8n^2}\varepsilon^3 + O\left(\varepsilon^4, n^{-3}\right),$$

$$\Delta_{D,1+} = 3 - \varepsilon + \frac{m}{4n}\varepsilon^2 - \frac{m(m+3)}{2n^2}\varepsilon^2 - \frac{m}{16n}\varepsilon^3 + \frac{7m(m+3)}{8n^2}\varepsilon^3 + O\left(\varepsilon^4, n^{-3}\right),$$

$$\Delta_{C,1-} = 3 - \varepsilon + \frac{m+2}{4n}\left(1 - \frac{2(m^2 + 4m - 12)}{mn}\right)\varepsilon^2 - \frac{m+2}{16n}\left(1 - \frac{2(7m^2 + 22m - 80)}{mn}\right)\varepsilon^3 + O\left(\varepsilon^4, n^{-3}\right),$$

$$\Delta_{D,1-} = 3 - \varepsilon + \frac{m}{4n}\varepsilon^2 - \frac{m^2 + 3m - 12}{2n^2}\varepsilon^2 - \frac{m}{16n}\varepsilon^3 + \frac{7m^2 + 21m - 80}{8n^2}\varepsilon^3 + O\left(\varepsilon^4, n^{-3}\right), \quad (67)$$

and the central charges

$$\frac{C_{T+}}{C_{T,\text{free}}} = 1 - \frac{5(m+1)}{24n}\varepsilon^2 + \frac{5\left(2m^2 + 9m + 17\right)}{24n^2}\varepsilon^2$$
$$- \frac{7(m+1)}{72n}\varepsilon^3 - \frac{31m^2 + 117m + 196}{72n^2}\varepsilon^3 + O\left(\varepsilon^4, n^{-3}\right),$$

$$\frac{C_{T-}}{C_{T,\text{free}}} = 1 - \frac{5(m+2)(m-1)}{24mn}\left(1 - \frac{2m^2 + 7m - 22}{mn}\right)\varepsilon^2$$
$$- \frac{7(m+2)(m-1)}{72mn}\left(1 + \frac{31m^2 + 86m - 356}{7mn}\right)\varepsilon^3 + O\left(\varepsilon^4, n^{-3}\right).$$
$$(68)$$

More results can be found in the ancillary data file, as described in Appendix A.

## 4.2 Results at large $n$

As mentioned in the introduction, we can give a description at large $n$ by introducing Hubbard–Stratonovich operators in the $S$ and $W$ representations. This is in agreement with the results (64) and (65) in the $\varepsilon$ expansion above. As a starting point for the analytic bootstrap at large $n$ we will therefore assume that these two representations contain scalar operators $\mathcal{S}$ and $\mathcal{W}$, with dimensions and OPE coefficients given by

$$\Delta_{\mathcal{R}} = 2 + O\left(\frac{1}{n}\right), \qquad c_{\phi\phi\mathcal{R}}^2 = \frac{a_{\mathcal{R}}}{n} + O\left(\frac{1}{n^2}\right), \qquad \mathcal{R} = \mathcal{S}, \mathcal{W}. \quad (69)$$

We will take these representations to consitute group III in our implementation of the recipe of section 3.3. The next task is to determine what operators contribute at order $1/n$ to the broken currents in all of the nine reprensentations in (62). This is done by expanding the crossing matrix (63) at large $n$ and studying the relative scaling of elements in the first column, where $\mathbb{1}$ contributes, and the other columns. We identify that for representations in group $\text{I} = \{X, Y, Z, B, C, D\}$, only the auxiliary fields generate order $1/n$ corrections, whereas for group $\text{II} = \{S, W, A\}$ also the currents in group I need to be taken into account.

Having identified the groups I, II and III, we follow the implementation of section 3.3 and generate CFT-data at order $1/n$ in all representations. In particular, the scaling dimensions are

given by

$$\Delta_{R,\ell} = 2(\mu - 1) + \ell + \frac{2\gamma_\phi^{(1)}}{n} + \gamma_{R,\ell} + O\left(\frac{1}{n^2}\right),\tag{70}$$

for $\gamma_{R,\ell}$ given by (55) and (57) and $\mu = d/2$. These expressions depend on three undetermined constants, $a_S$, $a_W$ and $\gamma_\phi^{(1)}$. Fortunately, the stress-energy tensor and the two global symmetry currents $J_A^\mu$ and $J_B^\mu$ provide three consistency equations for these unknowns, namely

$$\Delta_{S,2} = d, \qquad \Delta_{A,1} = d - 1, \qquad \Delta_{B,1} = d - 1.\tag{71}$$

There are four solutions to these equations, which we can identify with the four fixed points of the $\varepsilon$ expansion,

$$
\begin{aligned}
\text{Free:} \quad & a_S = 0, & a_W &= 0, & \gamma_\phi^{(1)} &= 0, \\[2mm]
O(mn): \quad & a_S = \frac{\mu(\mu-1)}{(2-\mu)^2}\frac{\eta_1^{O(N)}}{m}, & a_W &= 0, & \gamma_\phi^{(1)} &= \frac{\eta_1^{O(N)}}{m}, \\[2mm]
\text{Chiral:} \quad & a_S = \frac{\mu(\mu-1)\eta_1^{O(N)}}{m(2-\mu)^2}, & a_W &= \frac{\mu(\mu-1)\eta_1^{O(N)}}{2(2-\mu)^2}, & \gamma_\phi^{(1)} &= \frac{(m+1)\eta_1^{O(N)}}{2}, \\[2mm]
\text{Antichiral:} \quad & a_S = 0, & a_W &= \frac{\mu(\mu-1)\eta_1^{O(N)}}{2(2-\mu)^2}, & \gamma_\phi^{(1)} &= \frac{(m+2)(m-1)\eta_1^{O(N)}}{2m},
\end{aligned}\tag{72}
$$

where $\eta_1^{O(n)}$ is the anomalous dimension of $\phi$ in the $O(N)$ model, given in (60). The values for $\gamma_\phi^{(1)}$ are consistent with the literature results quoted in (8) and (9).

In Table 2 we summarize the twist families of the $O(m) \times O(n)$ symmetric theory at large $n$ in the chiral and antichiral fixed points. We give the leading twist family in each representation, and we also display a couple of subleading families in the singlet representation. The existence of each of these subleading families follows from the initial analytic bootstrap considerations of [33, 34], since they are the double-twist operators in a suitable four-point function. Importantly, these families contain more than one operator at each spin and therefore participate in mixing. In the cases where there is an operator at spin zero, we match it with the scalar singlets presented in Table 1.

In Table 2 we also explain how the scaling dimension of each twist family relates to the corresponding scalar. In similarity with the $O(N)$ model, we assume that the expressions (70) can be analytically continued to spin zero, giving the dimension or the shadow dimension of the corresponding scalar. Including also $\Delta_\phi$, this gives for the chiral fixed point

$$\Delta_{\phi+} = \mu - 1 + \frac{m+1}{2}\frac{\eta_1^{O(N)}}{n} + \ldots \qquad \overset{3d}{=} \frac{1}{2} + \frac{2(m+1)}{3\pi^2 n} + \ldots,$$

$$\Delta_{S+} = 2 - \frac{2(\mu-1)(2\mu-1)(m+1)}{2-\mu}\frac{\eta_1^{O(N)}}{n} + \ldots \qquad \overset{3d}{=} 2 - \frac{16(m+1)}{3\pi^2 n} + \ldots,$$

$$\Delta_{W+} = 2 + \left(\frac{2(m+3)}{\mu-2} + 2\mu(m+2) + 2\right)\frac{\eta_1^{O(N)}}{n} + \ldots \qquad \overset{3d}{=} 2 - \frac{4(m+4)}{3\pi^2 n} + \ldots,$$

$$\Delta_{X+} = 2\Delta_{\phi+} + \frac{\mu(m+1)}{2-\mu}\frac{\eta_1^{O(N)}}{n} + \ldots \qquad \overset{3d}{=} 1 + \frac{16(m+1)}{3\pi^2 n} + \ldots,$$

$$\Delta_{Y+} = 2\Delta_{\phi+} + \frac{\mu}{2-\mu}\frac{\eta_1^{O(N)}}{n} + \ldots \qquad \overset{3d}{=} 1 + \frac{4(m+4)}{3\pi^2 n} + \ldots,$$

$$\Delta_{Z+} = 2\Delta_{\phi+} - \frac{\mu}{2-\mu}\frac{\eta_1^{O(N)}}{n} + \dots \qquad \overset{3d}{=} 1 + \frac{4(m-2)}{3\pi^2 n} + \dots, \qquad (73)$$

and for the antichiral fixed point

$$\Delta_{\phi-} = \mu - 1 + \frac{(m+2)(m-1)}{2m}\frac{\eta_1^{O(N)}}{n} + \dots \qquad \overset{3d}{=} \frac{1}{2} + \frac{2(m+2)(m-1)}{3\pi^2 mn} + \dots,$$

$$\Delta_{S-} = 2\Delta_{\phi-} + \frac{\mu(4\mu-5)(m-1)(m+2)}{(2-\mu)m}\frac{\eta_1^{O(N)}}{n} + \dots \qquad \overset{3d}{=} 1 + \frac{16(m-1)(m+2)}{3\pi^2 mn} + \dots,$$

$$\Delta_{W-} = 2 + \left((1+2\mu)(m-4) + \frac{m^2+3m-10}{\mu-2} + m^2\mu\right)\frac{2\eta_1^{O(N)}}{mn} + \dots \qquad \overset{3d}{=} 2 - \frac{4(m^2+4m-8)}{3\pi^2 mn} + \dots,$$

$$\Delta_{X-} = 2\Delta_{\phi-} + \frac{\mu(m-1)(m+2)}{(2-\mu)m}\frac{\eta_1^{O(N)}}{n} + \dots \qquad \overset{3d}{=} 1 + \frac{16(m-1)(m+2)}{3\pi^2 mn} + \dots,$$

$$\Delta_{Y-} = 2\Delta_{\phi-} + \frac{\mu(m-2)}{(2-\mu)m}\frac{\eta_1^{O(N)}}{n} + \dots \qquad \overset{3d}{=} 1 + \frac{4(m^2+4m-8)}{3\pi^2 mn} + \dots,$$

$$\Delta_{Z-} = 2\Delta_{\phi-} - \frac{\mu(m+2)}{(2-\mu)m}\frac{\eta_1^{O(N)}}{n} + \dots \qquad \overset{3d}{=} 1 + \frac{4(m-4)(m+2)}{3\pi^2 mn} + \dots. \qquad (74)$$

The values for $\phi$, $S$, $W$ and $Z$ agree with those quoted in section 2.2, whereas we are not aware of any previous results for the remaining operators. We also give results for the non-conserved spin one operators

$$\Delta_{C,1+} = 2\Delta_{\phi+} + 1 + \frac{\eta_1^{O(N)}}{n} + \dots \qquad \overset{3d}{=} 2 + \frac{4(m+2)}{3\pi^2 n} + \dots,$$

$$\Delta_{D,1+} = 2\Delta_{\phi+} + 1 - \frac{\eta_1^{O(N)}}{n} + \dots \qquad \overset{3d}{=} 2 + \frac{4m}{3\pi^2 n} + \dots,$$

$$\Delta_{C,1-} = 2\Delta_{\phi-} + 1 + \frac{(m+2)\eta_1^{O(N)}}{mn} + \dots \qquad \overset{3d}{=} 2 + \frac{4(m+2)}{3\pi^2 n} + \dots,$$

$$\Delta_{D,1-} = 2\Delta_{\phi-} + 1 - \frac{(m-2)\eta_1^{O(N)}}{mn} + \dots \qquad \overset{3d}{=} 2 + \frac{4m}{3\pi^2 n} + \dots. \qquad (75)$$

The computation of the OPE coefficients provides results for the central charges, by (17). For the chiral fixed point we get

$$\frac{C_{T+}}{C_{T,\text{free}}} = 1 - \frac{(m+1)c_1}{n} + \dots \qquad \overset{3d}{=} 1 - \frac{20(m+1)}{9\pi^2 n} + \dots,$$

$$\frac{C_{J_A+}}{C_{J_A,\text{free}}} = 1 + \frac{c_2}{n} - \frac{(m+2)c_3}{2n} + \dots \qquad \overset{3d}{=} 1 - \frac{44+38m}{9\pi^2 n} + \dots,$$

$$\frac{C_{J_B+}}{C_{J_B,\text{free}}} = 1 - \frac{(m+1)c_2}{n} + \dots \qquad \overset{3d}{=} 1 - \frac{32(m+1)}{9\pi^2 n} + \dots, \qquad (76)$$

and for the antichiral fixed point

$$\frac{C_{T-}}{C_{T,\text{free}}} = 1 - \frac{(m+2)(m-1)c_1}{mn} + \dots \qquad \overset{3d}{=} 1 - \frac{20(m+2)(m-1)}{9\pi^2 mn} + \dots,$$

$$\frac{C_{J_A-}}{C_{J_A,\text{free}}} = 1 + \frac{(m+2)c_2}{mn} - \frac{(m+2)c_3}{2n} + \dots \qquad \overset{3d}{=} 1 - \frac{2(m+2)(19m-16)}{9\pi^2 mn} + \dots,$$

$$\frac{C_{J_B-}}{C_{J_B,\text{free}}} = 1 - \frac{(m+2)(m-1)c_2}{mn} + \dots \quad\overset{\text{3d}}{=} 1 - \frac{32(m+2)(m-1)}{9\pi^2 mn} + \dots, \quad (77)$$

where the precise form of the constants $c_i$ is given in (87) in Appendix B.

Table 2: Twist families in the large-$n$ expansion. We give a couple of subleading twist families in the singlet case, and the leading family in the other irreps. We denote degenerate operators by $\langle \cdot \rangle$.

| | | Chiral | | | | Antichiral | | |
|---|---|---|---|---|---|---|---|---|
| $R$ | Gp | $\mathcal{O}_\ell$ | $\tau_\infty$ | constraints | Gp | $\mathcal{O}_\ell$ | $\tau_\infty$ | constraints |
| $S$ | III | $\mathcal{J}_{S,\ell}$ | $2\Delta_\phi$ | $\begin{cases}\Delta_0 = d - \Delta_S \\ \Delta_2 = d\end{cases}$ | II | $\mathcal{J}_{S,\ell}$ | $2\Delta_\phi$ | $\begin{cases}\Delta_0 = \Delta_S \\ \Delta_2 = d\end{cases}$ |
| | — | $\langle [\phi,\phi]_{S,1,\ell} \rangle$ | $2\Delta_\phi + 2$ | $\ell \geqslant 2$ | — | $\langle \partial^\ell \phi_S^4 \rangle$ | $4\Delta_\phi$ | $\Delta_0 = \Delta_{S'}$ |
| | — | $\left\langle \begin{matrix}[S,S]_{S,0,\ell} \\ [\mathcal{W},\mathcal{W}]_{S,0,\ell}\end{matrix} \right\rangle$ | $4$ | $\Delta_0 = \langle \Delta_{S'}, \Delta_{S''} \rangle$ | — | $\langle [\phi,\phi]_{S,1,\ell} \rangle$ | $2\Delta_\phi + 2$ | $\ell \geqslant 2$ |
| $W$ | III | $\mathcal{J}_{W,\ell}$ | $2\Delta_\phi$ | $\Delta_0 = d - \Delta_W$ | III | $\mathcal{J}_{W,\ell}$ | $2\Delta_\phi$ | $\Delta_0 = d - \Delta_W$ |
| $X$ | I | $\mathcal{J}_{X,\ell}$ | $2\Delta_\phi$ | $\Delta_0 = \Delta_X$ | I | $\mathcal{J}_{Z,\ell}$ | $2\Delta_\phi$ | $\Delta_0 = \Delta_X$ |
| $Y$ | I | $\mathcal{J}_{Y,\ell}$ | $2\Delta_\phi$ | $\Delta_0 = \Delta_Y$ | I | $\mathcal{J}_{Y,\ell}$ | $2\Delta_\phi$ | $\Delta_0 = \Delta_Y$ |
| $Z$ | I | $\mathcal{J}_{Z,\ell}$ | $2\Delta_\phi$ | $\Delta_0 = \Delta_Z$ | I | $\mathcal{J}_{Z,\ell}$ | $2\Delta_\phi$ | $\Delta_0 = \Delta_Z$ |
| $A$ | II | $\mathcal{J}_{A,\ell}$ | $2\Delta_\phi$ | $\Delta_1 = d - 1$ | II | $\mathcal{J}_{A,\ell}$ | $2\Delta_\phi$ | $\Delta_1 = d - 1$ |
| $B$ | I | $\mathcal{J}_{B,\ell}$ | $2\Delta_\phi$ | $\Delta_1 = d - 1$ | I | $\mathcal{J}_{B,\ell}$ | $2\Delta_\phi$ | $\Delta_1 = d - 1$ |
| $C$ | I | $\mathcal{J}_{C,\ell}$ | $2\Delta_\phi$ | | I | $\mathcal{J}_{C,\ell}$ | $2\Delta_\phi$ | |
| $D$ | I | $\mathcal{J}_{D,\ell}$ | $2\Delta_\phi$ | | I | $\mathcal{J}_{D,\ell}$ | $2\Delta_\phi$ | |

# 5 Numerical bootstrap of $O(m) \times O(n)$ CFTs

## 5.1 Single correlator

In the single-correlator bootstrap, for which the crossing equations are discussed in Appendix C, we have obtained bounds on the dimensions of the leading scalar operators in the representations $S, W, X, Y, Z$ as functions of the dimension of $\phi$. The most interesting results are obtained in the $W$ and $X$ plots. More specifically, using large-$n$ results we can see that the (mild) kinks that appear in the $X$-bounds (see Fig. 2) are due to the chiral fixed points, while the kinks that appear in the $W$ bounds (see Fig. 3) are due to the antichiral fixed points. While this is clear at large $n$ only, we will assign the same meaning to the kinks at smaller $n$, but only down to $n = 6$ where, as we will see below, the situation becomes more subtle. As seen in Fig. 3, obtained with $m = 2$ for different values of $n$, there are very sharp kinks at large $n$ that get smoothed out as $n$ decreases. In Fig. 2 the kinks are much milder. They certainly exist at large $n$, but they are not so clear at low $n$—see Fig. 4—even when at the same $n$, e.g. $n = 4$, there is a clear kink in Fig. 3.

At $n = 6$ we can see from Fig. 5 that kinks exist in both bounds, although the one in the $\Delta_X$ bound is quite mild. While a hint of a kink in the $\Delta_X$ bound of the $O_{2,5}$ theory exists, the $\Delta_X$ bounds in the $O_{2,4}$ and $O_{2,3}$ theories are very smooth, although a change in slope can still be seen. These considerations suggest that the chiral fixed point ceases to exist for some $n$ between 5 and 6. This is a very rough estimate based on qualitative features of the $\Delta_X$ bounds. A more accurate estimate cannot be made based on the presence or absence of kinks as described here.

The persistence of the kinks in the $\Delta_W$ bounds even at small $n$ ($n = 4, 5$ and $n = 3$ although the kink is much milder for $n = 3$), combined with their absence in the $\Delta_X$ bounds, is rather puzzling. After all, intuition from the $\varepsilon$ expansion dictates that the antichiral and chiral fixed points, to which we have attributed the kinks in the $\Delta_W$ and $\Delta_X$ bounds, respectively, annihilate and become complex fixed points at some $n$ (in this case $n^+(2)$, whose estimated

value in the $\varepsilon$ expansion is 5.96(19) [27]). Since the bootstrap excludes nonunitary theories, both kinks are expected to disappear at some $n$ around 6, and indeed this is borne out to some extent for the kinks in the $\Delta_X$ bounds, as we discussed in the previous paragraph.

One explanation for the persistence of the $\Delta_W$ kinks at small $n$ is that our numerical bounds are insensitive to the putative nonunitarity of the antichiral fixed point for small $n$. This scenario could be further examined by estimating the size of this nonunitarity in perturbation theory, in a properly quantified sense that we do not discuss here, and comparing it with that of the chiral fixed point. We do not pursue this direction here, but it is worth investigating in the future. Another possibility is that the kinks in the $\Delta_W$ bounds at small $n$ are due to another fixed point, which is not the naive continuation of the antichiral fixed point to small $n$. Evidence for the existence of such a fixed point, belonging to the chiral universality class, exists in the literature; see [14, Sec. 11.5.3] and references therein. A further universality class, typically called collinear, is supposed to exist for small $n$.[12] These chiral and collinear fixed points arise after resummations of the perturbative beta functions. However, this approach has been criticized in [15], and the functional RG predicts that they do not exist [16, 17]. In section 5.3 below we will see that, consistently with the conclusions of [43], the chiral and collinear universality classes appear to exist for $O(2) \times O(3)$ CFTs, although we are not able to conclusively exclude small unitarity violations.

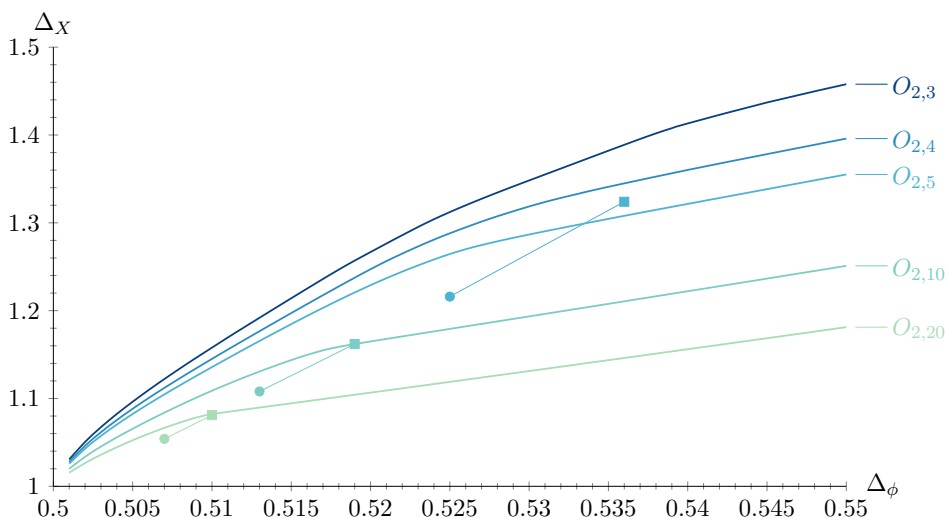

Figure 2: Upper bound on the dimension of the first scalar $X$ operator in the $\phi_{ar} \times \phi_{bs}$ OPE as a function of the dimension of $\phi$. Areas above the curves are excluded in the corresponding theories. The positions of the fixed points as predicted by the large-$n$ results (8), (9) for $\Delta_\phi$ and (73), (74) for $\Delta_X$ for $n = 5, 10, 20$ are also given as squares and circles for the chiral and antichiral fixed points, respectively. (The lines between squares and circles are added to illustrate fixed points with the same symmetry.)

---

[12]The $O(2) \times O(3)$ and $O(2) \times O(4)$ theories were also studied with the numerical bootstrap in [43], where a discussion of the chiral and collinear fixed points of the $O(2) \times O(3)$ and $O(2) \times O(4)$ theories can also be found. Our numerical bounds are consistent with those of [43].

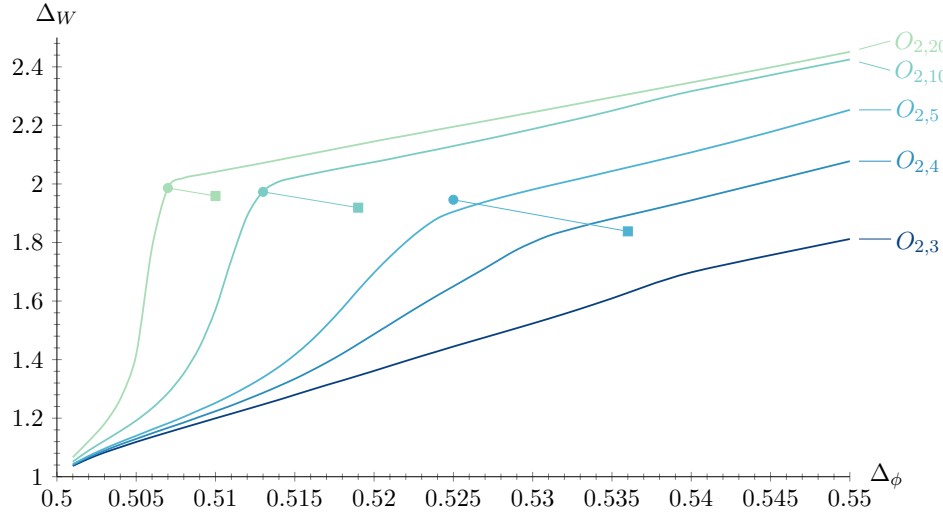

Figure 3: Upper bound on the dimension of the first scalar $W$ operator in the $\phi_{ar} \times \phi_{bs}$ OPE as a function of the dimension of $\phi$. Areas above the curves are excluded in the corresponding theories. The positions of the fixed points as predicted by the large-$n$ results (8) and (9) for $n = 5, 10, 20$ are also given as squares and circles for the chiral and antichiral fixed points, respectively. (The lines between squares and circles are added to illustrate fixed points with the same symmetry.)

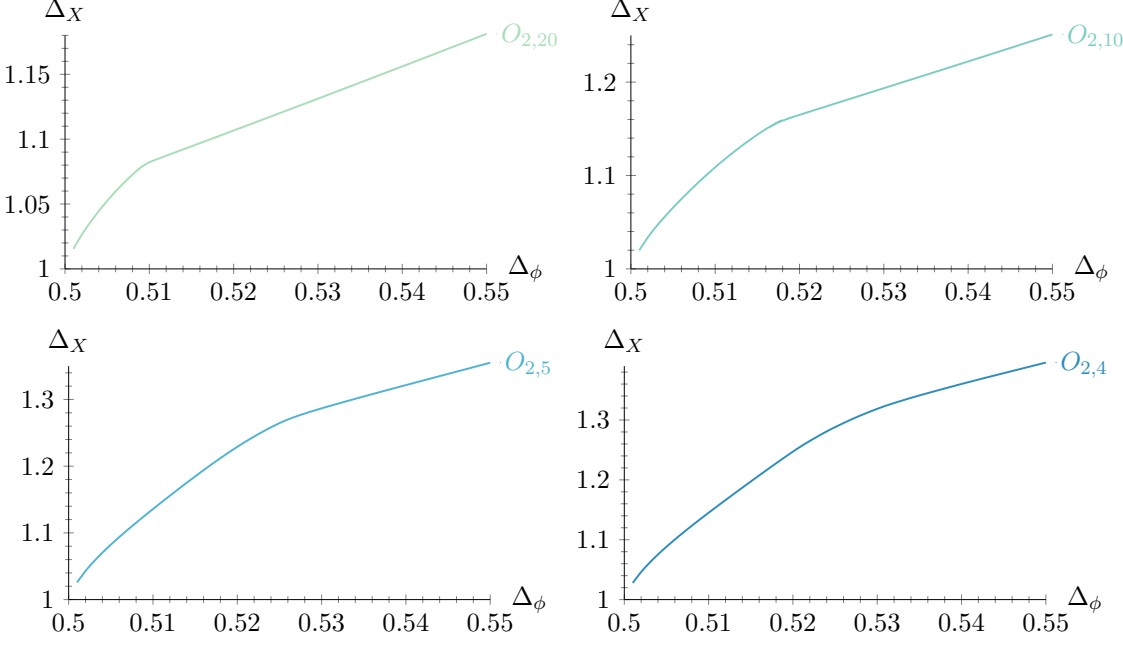

Figure 4: Upper bound on the dimension of the first scalar $X$ operator in the $\phi_{ar} \times \phi_{bs}$ OPE as a function of the dimension of $\phi$. Areas above the curves are excluded in the corresponding theories.

## 5.2 Mixed correlators

In the mixed correlator system, to find an island around the chiral fixed point we use the following assumptions (both for the $O_{2,10}$ and the $O_{2,20}$ theory):

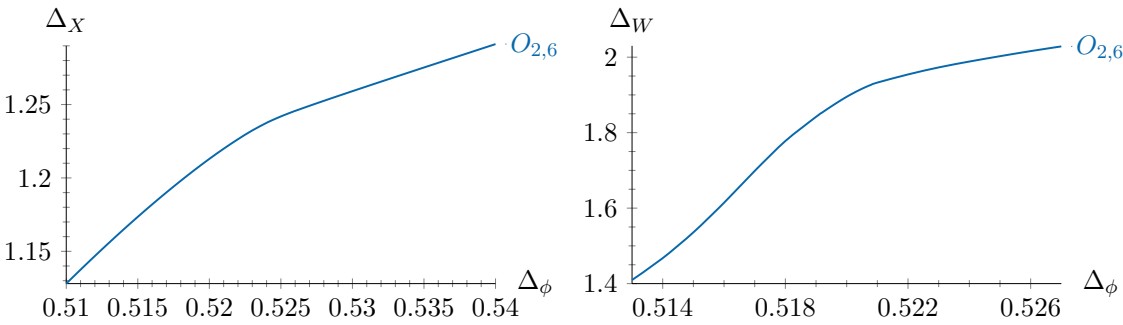

Figure 5: Upper bound on the dimension of the first scalar $X$ and the first scalar $W$ operator in the $\phi_{ar} \times \phi_{bs}$ OPE as a function of the dimension of $\phi$ in the $O_{2,6}$ theory.

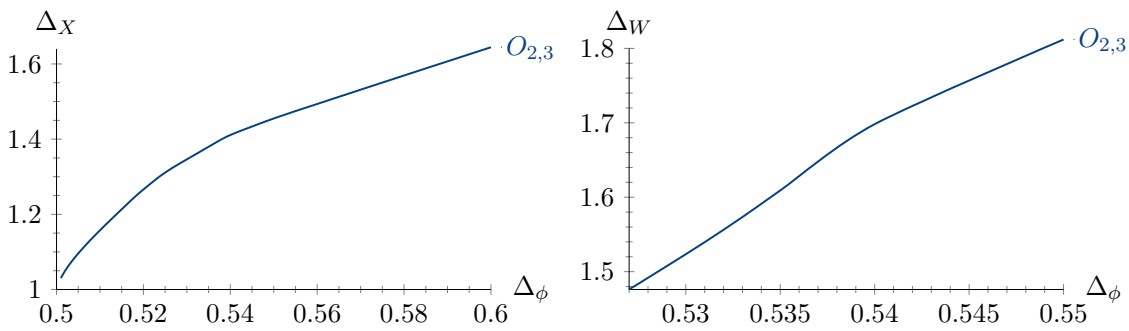

Figure 6: Upper bound on the dimension of the first scalar $X$ and the first scalar $W$ operator in the $\phi_{ar} \times \phi_{bs}$ OPE as a function of the dimension of $\phi$ in the $O_{2,3}$ theory.

(C-1) saturation of $X$ bound of Fig. 2,

(C-2) existence of conserved current in the $B$ sector, i.e. $\Delta_{J_B^\mu} = 2$,

(C-3) dimension of next-to-leading vector operator in the $B$ sector, $J_B^{\mu\prime}$, above 3, i.e. $\Delta_{J_B^{\mu\prime}} \geqslant 3$,

(C-4) dimension of next-to-leading scalar singlet, $S'$, above 3, i.e. $\Delta_{S'} \geqslant 3$,

(C-5) dimension of next-to-leading bifundamental operator, $\phi'$, allowed slightly above $\Delta_\phi$, i.e. $\Delta_{\phi'} \geqslant \Delta_\phi + 0.01$.

For the antichiral fixed point of the $O_{2,20}$ theory we make the following assumptions:

(A-$O_{2,20}$-1) saturation of $W$ bound of Fig. 3,

(A-$O_{2,20}$-2) existence of conserved current in the $B$ sector, i.e. $\Delta_{J_B^\mu} = 2$,

(A-$O_{2,20}$-3) dimension of next-to-leading vector operator in the $B$ sector, $J_B^{\mu\prime}$, above 3, i.e. $\Delta_{J_B^{\mu\prime}} \geqslant 3$,

(A-$O_{2,20}$-4) dimension of next-to-leading scalar singlet, $S'$, above 1.5, i.e. $\Delta_{S'} \geqslant 1.5$.

(A-$O_{2,20}$-5) dimension of next-to-leading bifundamental operator, $\phi'$, allowed slightly above $\Delta_\phi$, i.e. $\Delta_{\phi'} \geqslant \Delta_\phi + 0.01$.

Finally, for the antichiral fixed point of the $O_{2,10}$ theory we make the following assumptions:

(A-$O_{2,10}$-1)  saturation of $W$ bound of Fig. 3,

(A-$O_{2,10}$-2)  existence of conserved current in the $B$ sector, i.e. $\Delta_{J_B^\mu} = 2$,

(A-$O_{2,10}$-3)  dimension of next-to-leading vector operator in the $B$ sector, $J_B^{\mu\prime}$, above 3, i.e. $\Delta_{J_B^{\mu\prime}} \geqslant 3$,

(A-$O_{2,10}$-4)  dimension of next-to-leading scalar singlet, $S'$, above 1.6, i.e. $\Delta_{S'} \geqslant 1.6$.

(A-$O_{2,10}$-5)  dimension of next-to-leading bifundamental operator, $\phi'$, above 1.6, i.e. $\Delta_{\phi'} \geqslant 1.6$.

Let us note here that even with $\Delta_{S'} \geqslant 3$ we obtain islands around antichiral fixed points, so long as we keep the gap on $\Delta_{\phi'}$ small. This is inconsistent with the fact that the antichiral fixed point is unstable, but with our numerical power (see Appendix C) we cannot see the inconsistency. However, when we increase the gap on $\Delta_{\phi'}$ sufficiently, we do see that the antichiral island disappears with $\Delta_{S'} \geqslant 3$. This is presumably due to the crossing equations that arise from the $\langle \phi\phi SS \rangle$ four-point function, which in the $12 \to 34$ channel are sensitive to both $\phi'$ and $S$, while in the $14 \to 32$ channel they are sensitive to $\phi'$ but not $S$.

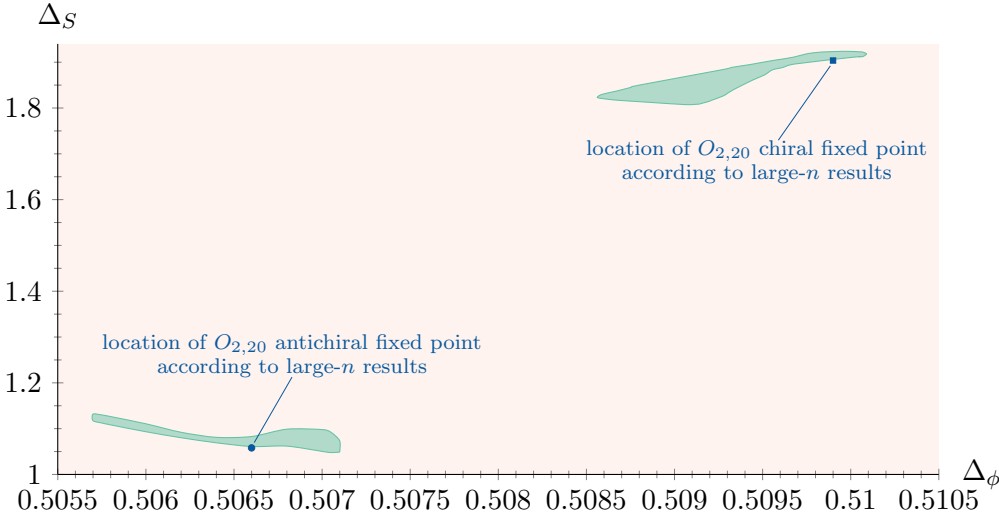

Figure 7: Allowed region (in green) for the $O_{2,20}$ chiral and antichiral fixed points and their location according to (8). The chiral island is obtained with the assumptions (C-1)-(C-5), while the antichiral island is obtained with the assumptions (A-$O_{2,20}$-1)-(A-$O_{2,20}$-5).

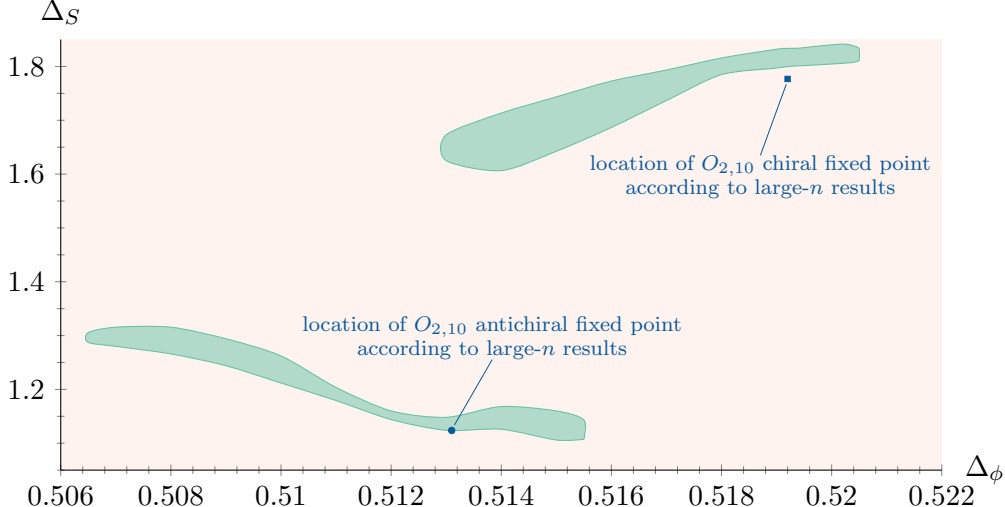

Figure 8: Allowed region (in green) for the $O_{2,10}$ chiral and antichiral fixed points and their location according to (9). The chiral island is obtained with the assumptions (C-1)-(C-5), while the antichiral island is obtained with the assumptions (A-$O_{2,10}$-1)-(A-$O_{2,10}$-5).

As we have already mentioned, the $\varepsilon$ expansion predicts that $n^+(2) = 5.96(19)$, meaning that a unitary chiral fixed point may exist for $n = 6$. The state-of-the-art analysis of the $O_{2,6}$ theory with the $\varepsilon$ expansion was performed recently in [27]. It turns out that we can also find an island with our nonperturbative numerical bootstrap methods, and so we can compare our results with large-$n$ and $\varepsilon$ expansion results. This is done in Fig. 9.

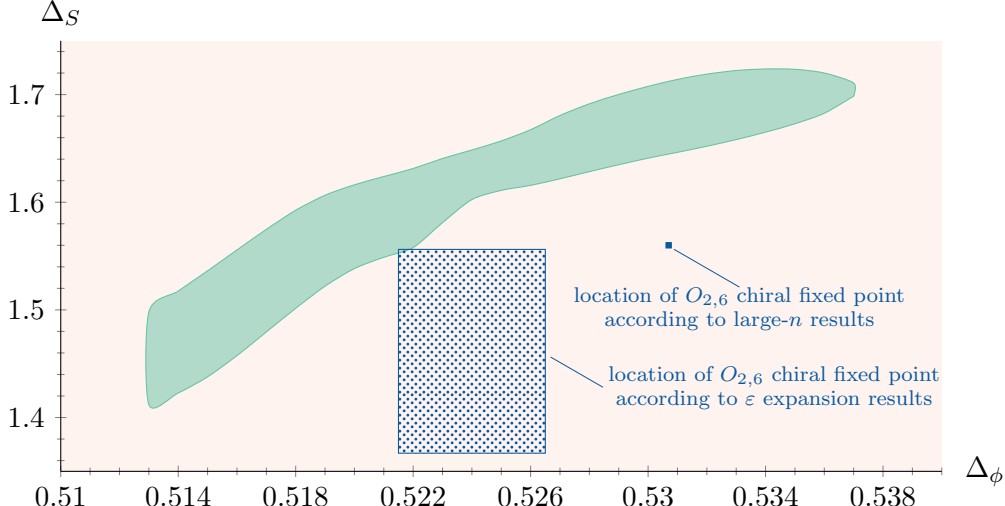

Figure 9: Allowed region (in green) for the $O_{2,6}$ chiral fixed point and its location according to (8) and the $\varepsilon$ expansion results of [27, Table 9]. This island is obtained with the assumptions (C-1)-(C-5) using saturation of the $O_{2,6}$ $X$ bound in Fig. 5.

### 5.3 Mixed correlators in the $O(2) \times O(3)$ case

The case $m = 2, n = 3$ is of particular interest since it is believed to appear as a symmetry in the continuum limit of frustrated spin models at criticality. We remind that the fixed points for $m = 2, n = 3$ arise after resummations of the perturbative beta functions, i.e. they are not

found in the standard perturbative $\varepsilon$ expansion. There are two fixed points, called chiral and collinear, with the chiral being of relevance for the experimentally observed phase transitions in stacked triangular antiferromagnets [14, Sec. 11.5], and the collinear potentially relevant for the normal-to-planar superfluid transition in ${}^3$He [55]. Monte Carlo simulations support the existence of these fixed points, but functional RG methods come to a different conclusion, namely that these fixed points do not exist and that experiments in frustrated magnets are actually seeing weakly first-order phase transitions [15–17].

In what follows we present results regarding putative theories that live on the $W$ and $Z$ sector single correlator bounds. The $W$ (resp. $Z$) sector bound has a mild kink that appears to correspond to the chiral (resp. collinear) fixed point. This observation was first made in [43]. Here we take the next logical step and perform a mixed correlator bootstrap around these kinks. We also compare to theoretical predictions and experimental data where applicable. We note that the general trend of sensitivity to assumptions in the $B$ sector persists here as well.

Let us start by noticing that, however mild, there seems to be a kink on the $W$ sector single correlator plot of Fig. 6 around $\Delta_\phi = 0.539$. The kink can be seen more clearly in Fig. 10. As mentioned above, evidence for the existence of the $O_{2,3}$ chiral fixed point has appeared before in the literature, based on resummations of perturbative beta functions. Such computations have been performed at six loops using the massive zero momentum (MZM) scheme [54] and at five loops using the modified minimal subtraction ($\overline{\text{MS}}$) scheme [52] (see also [51]). Monte Carlo simulations have also shown signs of a critical theory, with the most recent analysis performed in [56]. According to results of [53, Eq. (2.9)] and [51, Table III], the theory at the chiral fixed point has $\Delta_\phi = 0.545(20)$ and $\Delta_W = 1.79(9)$ in the $\overline{\text{MS}}$ scheme.[13] In the MZM scheme, [54, Table III] and [51, Table III] give $\Delta_\phi = 0.55(5)$ and $\Delta_W = 1.91(5)$. The $\overline{\text{MS}}$ scheme result is more consistent with the location of the kink in Fig. 10.

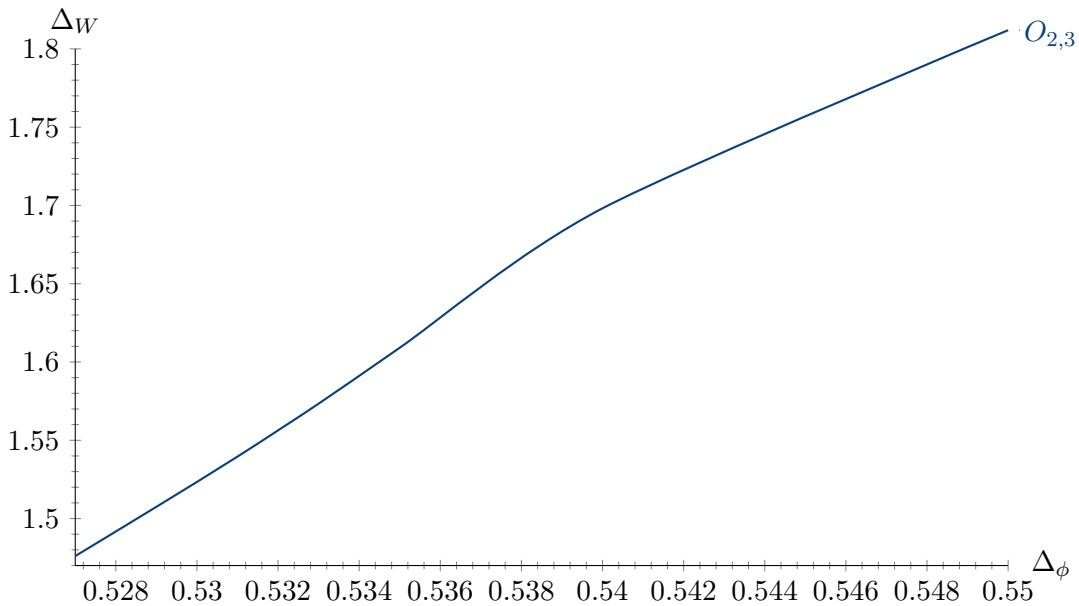

Figure 10: Upper bound on the dimension of the first scalar $W$ operator in the $\phi_{ar} \times \phi_{bs}$ OPE as a function of the dimension of $\phi$. The area above the curve is excluded.

If we assume

---

[13]In [51, Table III], $y_4 = 3 - \Delta_W$.

(C-$O_{2,3}$-1) saturation of $W$ bound of Fig. 10,

(C-$O_{2,3}$-2) existence of a conserved current in the $B$ sector, i.e. $\Delta_{J_B^\mu} = 2$,

(C-$O_{2,3}$-3) dimension of the next-to-leading vector operator in the $B$ sector, $J_B^{\mu\prime}$, above 2.4, i.e. $\Delta_{J_B^{\mu\prime}} \gtrsim 2.4$,

(C-$O_{2,3}$-4) dimension of the next-to-leading scalar singlet, $S'$, above 2, i.e. $\Delta_{S'} \gtrsim 2$.

(C-$O_{2,3}$-5) dimension of next-to-leading bifundamental operator, $\phi'$, above 1.5, i.e. $\Delta_{\phi'} \gtrsim 1.5$,

then we obtain Fig. 11. Let us note that the island in Fig. 11 remains even if we make the more constraining assumption $\Delta_{S'} \gtrsim 3$, which is compatible with the RG stability of the $O_{2,3}$ chiral fixed point. If instead of (C-$O_{2,3}$-3) we assume that $J_B^{\mu\prime}$ can appear with dimension below 2.4, then both the island and the peninsula are part of a bigger, continuous peninsula that includes both.

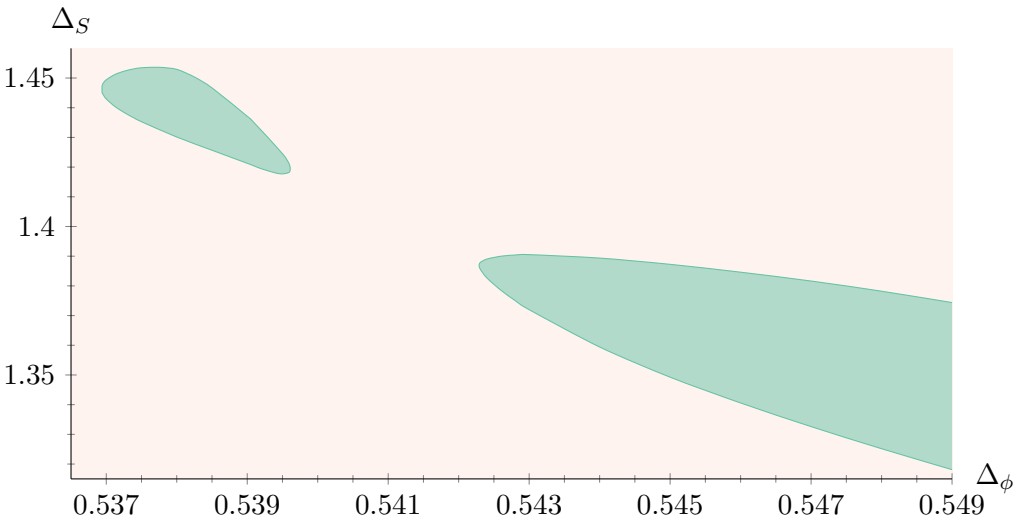

Figure 11: Allowed region (in green) for the $O_{2,3}$ chiral fixed point, obtained with the assumptions (C-$O_{2,3}$-1)-(C-$O_{2,3}$-5).

The state-of-the-art results in the literature for operator scaling dimensions of relevance for Fig. 11 are as follows:

$$[54]: \qquad \Delta_\phi = 0.55(5), \qquad \Delta_S = 1.18(10), \tag{78}$$

$$[53]: \qquad \Delta_\phi = 0.545(20), \quad \Delta_S = 1.41(13), \tag{79}$$

$$[56]: \qquad \Delta_\phi = 0.50(4), \qquad \Delta_S = 1.08(4). \tag{80}$$

Even with the large error bars in (80), we see that agreement is best with the results of [53], mainly due to $\Delta_S$.[14] In conjunction with the $\Delta_W$ results mentioned earlier, it is clear that the $\overline{\text{MS}}$ results of [53] and [51] agree best with our bootstrap results for the chiral fixed point.

Experimental results for transitions described by the $O_{2,3}$ chiral fixed point can be found in [14, Table 37]. The agreement of our results for the critical exponent $\nu = 1/(3-\Delta_S)$ is very good, although the same cannot be said for the critical exponent $\beta = \Delta_\phi/(3-\Delta_S)$.

---

[14]Note that the six-loop $\overline{\text{MS}}$ beta functions for $O(m) \times O(n)$ theories have recently been obtained in [27], so the five-loop analysis of [53] can perhaps be extended to six loops.

Refs. [53, 54, 56] indicate that their fixed points are of the focus type, which means that they are nonunitary, while our study pertains to unitary theories. It is unclear to us how sizable nonunitarities of the type discussed in [53, 54, 56] could have been missed by our bootstrap results. We note that the numerical bootstrap has previously found islands for theories that are believed to be nonunitary, namely the five-dimensional $O(N)$ models [69] and the Ising model in $d = 4 - \varepsilon$ [70]. In the former case, increasing the constraining power of the numerics led to the disappearance of the allowed region. It is possible that also our island in Fig. 11 will disappear with stronger numerics.

The $O_{2,3}$ collinear fixed point corresponds to a kink in the bound of the first scalar operators in the $Z$ irrep; see Fig. 12. According to results of [55] and [51, Table III], the theory at the collinear fixed point has $\Delta_\phi = 0.543(12)$ and $\Delta_Z = 1.8(1)$ in the $\overline{\text{MS}}$ scheme.[15] In the MZM scheme, [55] and [51, Table III] give $\Delta_\phi = 0.5395(35)$ and $\Delta_Z = 1.75(10)$. The consistency of the MZM scheme result with the location of the kink in Fig. 12 appears to be slightly better than that of the $\overline{\text{MS}}$ scheme result.

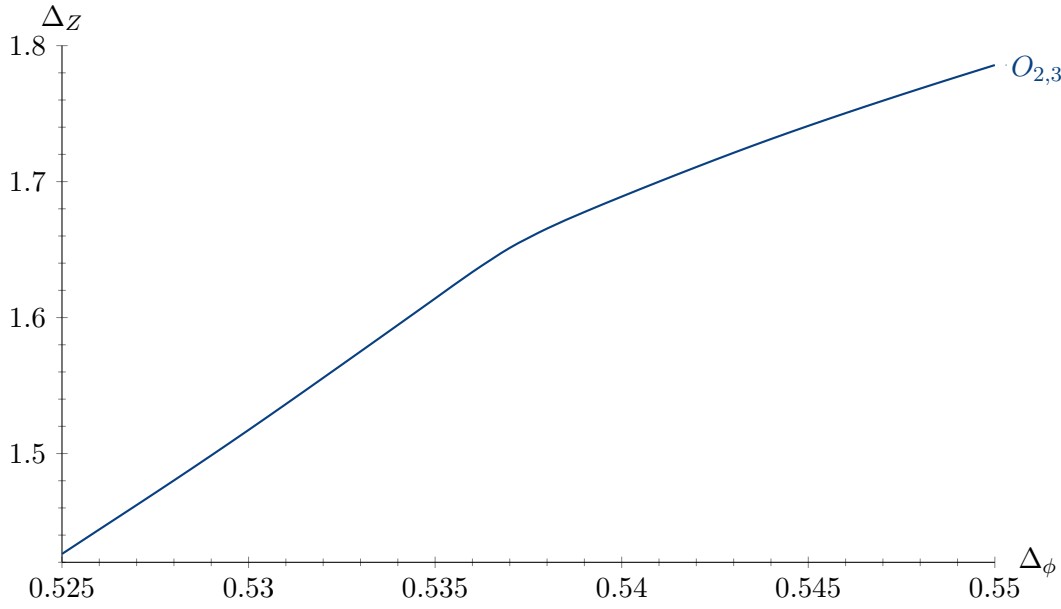

Figure 12: Upper bound on the dimension of the first scalar $Z$ operator in the $\phi_{ar} \times \phi_{bs}$ OPE as a function of the dimension of $\phi$. The area above the curve is excluded.

With the assumptions

(Coll-1)  saturation of $Z$ bound of Fig. 12,

(Coll-2)  existence of conserved current in the $B$ sector, i.e. $\Delta_{J_B^\mu} = 2$,

(Coll-3)  dimension of next-to-leading vector operator in the $B$ sector, $J_B^{\mu\prime}$, above 2.5, i.e. $\Delta_{J_B^{\mu\prime}} \geqslant 2.5$,

(Coll-4)  dimension of next-to-leading scalar singlet, $S'$, above 3, i.e. $\Delta_{S'} \geqslant 3$.

(Coll-5)  dimension of next-to-leading bifundamental operator, $\phi'$, slightly above $\Delta_\phi$, i.e. $\Delta_{\phi'} \geqslant \Delta_\phi + 0.01$,

---

[15]In [51, Table III], $y_1 = 3 - \Delta_Z$.

we find a rather large island that appears to end slightly to the left of the position of the kink in Fig. 12; see Fig. 13. If we weaken the assumption (Coll-3) to $\Delta_{J_B^{\mu\prime}} \geqslant 2.4$, then there is no separate island and peninsula, but rather a continuous peninsula that gets narrow in the region between the island and peninsula of Fig. 13. According to [55], in the $\overline{\text{MS}}$ scheme the $O_{2,3}$ collinear fixed point has $\Delta_\phi = 0.543(12)$ and $\Delta_S = 1.41(20)$, while in the MZM scheme it has $\Delta_\phi = 0.5395(35)$ and $\Delta_S = 1.31(11)$. Here the MZM scheme result for $\Delta_\phi$ and the $\overline{\text{MS}}$ scheme result for $\Delta_S$ appear to agree better with our island in Fig. 13. Just like the $O_{2,3}$ chiral fixed point analyzed above, it would be very interesting to study the effect of stronger numerics in this case as well.

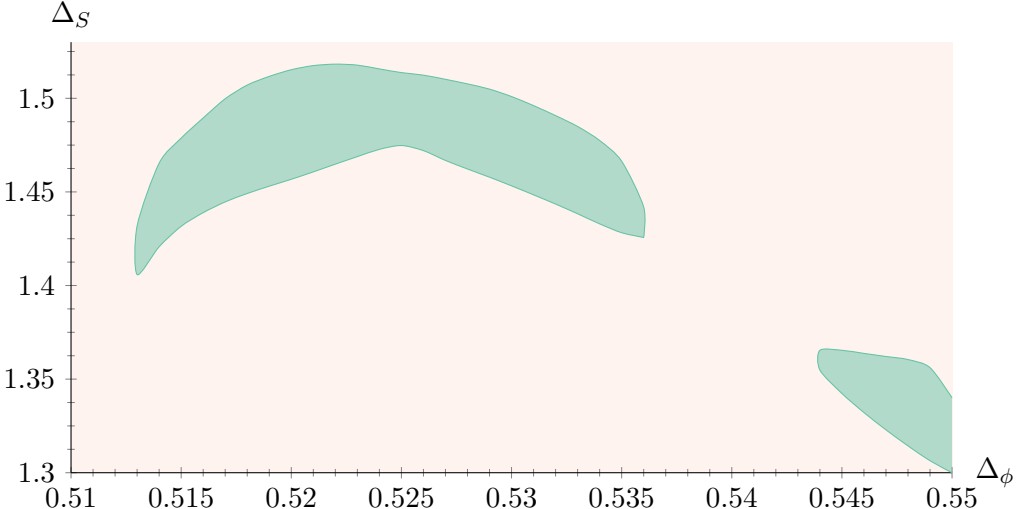

Figure 13: Allowed region (in green) for the $O_{2,3}$ collinear fixed point with the assumptions (Coll-1)-(Coll-5).

## 5.4 Single correlator in the $O(2) \times O(2)$ case

In the $O_{2,2}$ case, the collinear fixed point is equivalent to the $O(2)$ fixed point [51], and so we will only be examining the proposed chiral fixed point [51, 54]. As discussed in Appendix C, the single-correlator crossing equations of the $O_{2,2} = O(2)^2/\mathbb{Z}_2$ theory are equivalent to those of the $MN_{2,2} = O(2)^2 \rtimes S_2$ theory discussed in [39]. A strong kink was obtained in the $X$ sector of [39] (see Fig. 1 there), which corresponds to the $Z$ sector in (93) below. This bound is shown in Fig. 14. According to [54, Table III] and [51, Table III], in the MZM scheme the $O_{2,2}$ chiral fixed point has $\Delta_\phi = 0.54(5)$, $\Delta_S = 1.25(9)$, $\Delta_{WX} = 1.75(4)$, $\Delta_Y = 0.97(7)$ and $\Delta_Z = 0.46(12)$, while, in the $\overline{\text{MS}}$ scheme, [53, Eq. (2.8)] and [51, Table III] give $\Delta_\phi = 0.545(20)$, $\Delta_S = 1.46(14)$, $\Delta_{WX} = 1.66(15)$, $\Delta_Y = 1.00(15)$ and $\Delta_Z = 0.63(15)$.[16] Therefore, we conclude that the kink in Fig. 14 does not correspond to the $O_{2,2}$ chiral fixed point. In [39] it was suggested that this kink may correspond to the fully-interacting theory of the $\varepsilon$ expansion analyzed in [71–74].

---

[16]The notation $S, WX, Y, Z$ is explained in Appendix C.

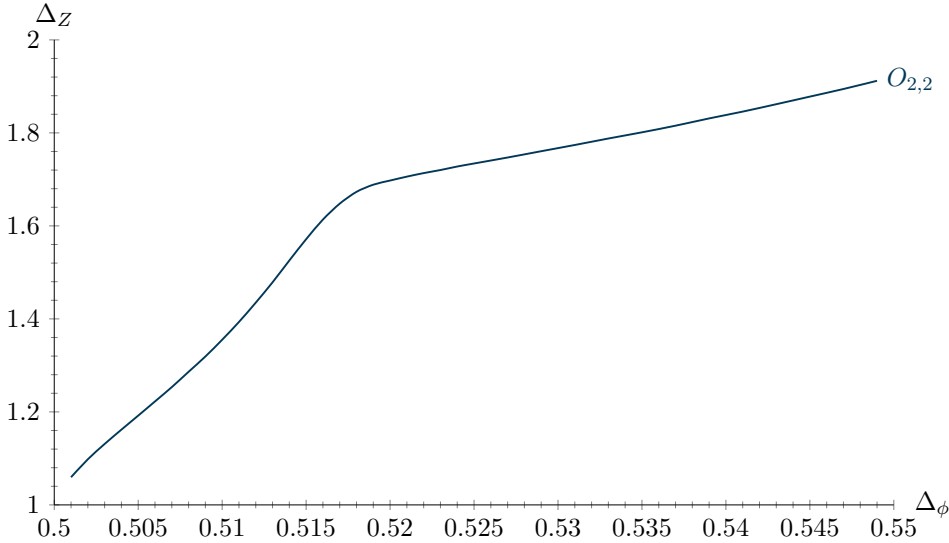

Figure 14: Upper bound on the dimension of the first scalar $Z$ operator in the $\phi_{ar} \times \phi_{bs}$ OPE as a function of the dimension of $\phi$. The area above the curve is excluded.

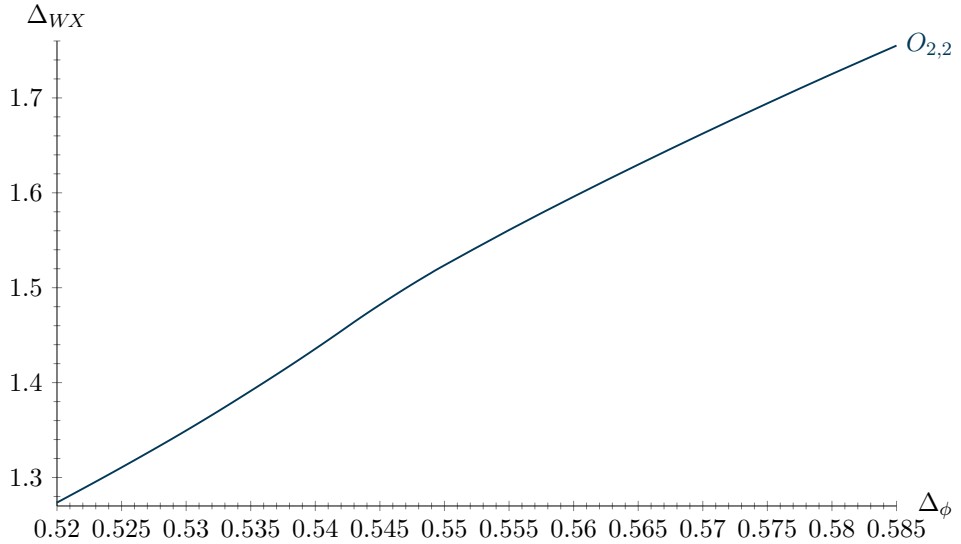

Figure 15: Upper bound on the dimension of the first scalar $WX$ operator in the $\phi_{ar} \times \phi_{bs}$ OPE as a function of the dimension of $\phi$. The area above the curve is excluded.

To continue our search for the $O_{2,2}$ chiral fixed point, we obtain a bound on the dimension of the first scalar operator in the $WX$ representation. The bound is shown in Fig. 15. An extremely mild change in the slope of the bound is observed at $\Delta_\phi = 0.547(2)$, at which point $\Delta_{WX} = 1.507(10)$. These numbers are in relatively good agreement with the $\overline{\text{MS}}$ scheme results mentioned in the previous paragraph. The mildness of the kink in Fig. 15, however, indicates that we need stronger numerics in order to reach solid conclusions. We leave this for future work.

# 6  Summary and conclusion

Conformal field theories with $O(m) \times O(n)$ global symmetry in $d = 3$ spacetime dimensions have generated intense interest and deep questions for many years. Standard methods like the $\varepsilon$ and large-$n$ expansions have been employed for their study, and have revealed a rather intricate structure of fixed points depending on the values of $m$ and $n$ (see Fig. 1). For example, it is widely believed that for $m = 2$ the large $n$ fixed points (chiral and antichiral) survive down to $n = 6$ as unitary fixed points, while for some $n$ between 5 and 6 they collide and become nonunitary for $n = 2, 3, 4, 5$. Similar results hold for $m > 2$ and corresponding values of $n$.

In this work we applied analytic and numerical bootstrap techniques to the study of $O(m) \times O(n)$ CFTs. Our analytic bootstrap results both corroborated and extended earlier results in the $\varepsilon$ and large-$n$ expansions. Indeed, we found the large-$n$ chiral and antichiral fixed points purely from analytic bootstrap considerations, and verified some of their properties as established with older methods. Our new analytic results consisted of $\varepsilon$ and large-$n$ expansion expressions for scaling dimensions of $\phi$-bilinears in all representations of operators that appear in the $\phi \times \phi$ OPE for spins zero and one, where $\phi$ transforms in the bifundamental representation of $O(m) \times O(n)$. These results allowed us to unequivocally identify kinks in our numerical bootstrap bounds with the known fixed points at large $n$. We also obtained analytic results for OPE coefficients related to the central charge $C_T$ and the coefficients $C_J$ of current-current two-point functions.

Our numerical bootstrap bounds were focused on the case $m = 2$. For $n = 10, 20$ we were able to find islands in the $(\Delta_\phi, \Delta_S)$ parameter space, where $\Delta_S$ is the dimension of the leading scalar singlet in the $\phi \times \phi$ OPE, corresponding to the chiral and antichiral fixed points and located in the region predicted by large-$n$ results (see Figs. 7 and 8). For the edge case of $m = 2, n = 6$ we also obtained an island (see Fig. 9). In all these cases we used a mixed correlator bootstrap. In the $m = 2, n = 6$ case we were able to compare our nonperturbative bootstrap results with the state-of-the-art resummed $\varepsilon$ expansion results of [27]. The agreement is reasonably good, but our island is rather large. We need stronger numerics and more refined bootstrap techniques in order to make our island smaller and obtain more accurate determinations of critical exponents.

CFTs with $O(2) \times O(3)$ and $O(2) \times O(2)$ global symmetry have been argued to have various applications to observed critical phenomena. However, when analyzed with different RG methods, notably resummations of perturbative beta functions, Monte Carlo and functional RG, the obtained results are not in mutual agreement. More specifically, beta function resummations [51–53] and Monte Carlo computations [56] indicate the presence of fixed points beyond the ones found with perturbative methods, while the functional RG concludes that such fixed points do not exist [15–17]. Correspondingly, there are two conflicting suggestions, namely that experiments are seeing critical (second-order) or weakly first-order behavior. It is important to note here that the beta function resummation and Monte Carlo fixed points are suggested to be nonunitary (focus type), with next-to-leading scalar singlets of complex scaling dimensions. We attempted to address these issues in this work. Our results provided support for the existence of these fixed points, but we saw no signs of nonunitarity.[17] Overall, we were unable to provide conclusive answers, but we believe that more dedicated bootstrap work with stronger numerics will be able to reach definitive conclusions in the near future.

Another area of interest concerns the chiral phase transition of two-flavor massless quantum chromodynamics [75] (with global symmetry $SU(2) \times SU(2)$) which has not been con-

---

[17]Ideally, assuming there is nonunitarity, we would expect the islands to disappear at some higher, a priori unknown, number of derivatives included in the bootstrap functional. Also, we emphasize that it is natural we do not see the nonunitarity in the analytic bootstrap since it does not capture the new fixed points found in resummations.

clusively demonstrated to be first- or second-order. A conformal bootstrap study of this question, as related to the three-dimensional $O(2) \times O(4)$ theory after assuming that the axial $U(1)$ symmetry is restored at the transition point so that the full global symmetry group is $U(1) \times SU(2) \times SU(2)$, has appeared in [43] (for a review see [76]). Based on the existence of kinks in bootstrap bounds, e.g. the one in Fig. 3, it was suggested in [43] that the phase transition is second order. The $O(2) \times O(4)$ case is similar to the $O(2) \times O(3)$ and $O(2) \times O(2)$ CFTs studied in this paper, in that the existence of the suggested fixed points has been questioned by other methods. We believe that with stronger numerics and perhaps larger mixed-correlator systems than those considered here it will be possible to settle this question with the conformal bootstrap.

We conclude with the Tables 3–5, which summarize our results of critical exponents for $O(2) \times O(3)$ and $O(2) \times O(2)$ theories, as well as corresponding results from the literature.[18]

Table 3: $O(2) \times O(3)$ "chiral" critical exponents.

| Method | $\alpha$ | $\beta$ | $\nu$ | $\gamma$ | $\delta$ | $\eta$ | $\phi_W$ |
|---|---|---|---|---|---|---|---|
| This work (Figs. 10, 11) | 0.082(22) | 0.344(5) | 0.639(7) | 1.23(3) | 4.573(14) | 0.077(3) | 0.818(16) |
| $\overline{\text{MS}}$ scheme [51,53] | 0.11(15) | 0.34(4) | 0.63(5) | 1.20(24) | 4.5(2) | 0.09(4) | 0.76(12) |
| MZM scheme [51,54] | 0.35(9) | 0.30(4) | 0.55(3) | 1.04(18) | 4.5(5) | 0.1(1) | 0.58(6) |
| Monte Carlo [56] | 0.44(3) | 0.26(3) | 0.52(1) | 1.04(9) | 5.0(5) | 0.00(8) | – |

Table 4: $O(2) \times O(3)$ "antichiral/collinear" critical exponents.

| Method | $\alpha$ | $\beta$ | $\nu$ | $\gamma$ | $\delta$ | $\eta$ | $\phi_Z$ |
|---|---|---|---|---|---|---|---|
| This work (Figs. 12, 13) | 0.05(7) | 0.341(19) | 0.650(23) | 1.27(11) | 4.72(13) | 0.049(23) | 0.89(4) |
| $\overline{\text{MS}}$ scheme [51,53] | 0.11(24) | 0.34(5) | 0.63(8) | 1.2(3) | 4.52(12) | 0.086(24) | 0.75(16) |
| MZM scheme [51,54] | 0.22(12) | 0.319(23) | 0.59(4) | 1.14(16) | 4.56(4) | 0.079(7) | 0.74(11) |

Table 5: $O(2) \times O(2)$ critical exponents.

| Method | $\alpha$ | $\beta$ | $\nu$ | $\gamma$ | $\delta$ | $\eta$ |
|---|---|---|---|---|---|---|
| Bootstrap (Fig. 14 and [39]) | 0.302(18) | 0.293(3) | 0.566(6) | 1.112(24) | 4.7952(21) | 0.0353(4) |
| $\varepsilon$ expansion [74] | −0.14(3) | 0.370(6) | 0.715(10) | 1.404(25) | 4.801(11) | 0.0343(20) |

Hints for the controversial chiral $O(2) \times O(2)$ model where found in Fig. 15, but we consider the strength of our numerics insufficient to provide estimates for critical exponents in this case.

## Acknowledgements

We are grateful to J. Gracey for comments on the manuscript and for pointing out some relevant literature. We would like to thank the organizers of "Bootstrap 2019" at Perimeter Institute where this work was initiated. Research at Perimeter Institute is supported in part by the

---

[18]The error bars in the first lines of Tables 3 and 4 correspond to the size of the islands, but are not rigorous due to the assumption that the theories must saturate the $\Delta_W$ and $\Delta_Z$ exclusion bounds, respectively. For the crossover exponents we use the values of $\Delta_W$ and $\Delta_Z$ that correspond to the positions where the slope changes in the corresponding exclusion bounds. Note also that Table 5 refers to the theory saturating the bound in Fig. 14; this theory is expected to be the one found in the $\varepsilon$ expansion. Lastly, the error bars for the numerical bootstrap in Table 14 are due to the use of the extremal functional method and are also not rigorous.

Government of Canada through the Department of Innovation, Science and Economic Development Canada and by the Province of Ontario through the Ministry of Economic Development, Job Creation and Trade. This research used resources provided by the Los Alamos National Laboratory Institutional Computing Program, which is supported by the U.S. Department of Energy National Nuclear Security Administration under Contract No. 89233218CNA000001. This research used resources of the National Energy Research Scientific Computing Center (NERSC), a U.S. Department of Energy Office of Science User Facility operated under Contract No. DE-AC02-05CH11231. Some numerical computations in this paper were run on the LX-PLUS cluster at CERN and the Metropolis cluster at the Crete Center for Quantum Complexity and Nanotechnology. AS is grateful for an allocation of computing time on the Caltech High Performance Cluster, provided by the Simons Foundation grant 48865 (Simons Collaboration on the Nonperturbative Bootstrap). The research work of SRK was supported by the Hellenic Foundation for Research and Innovation (HFRI) under the HFRI PhD Fellowship grant (Fellowship Number: 1026). Research presented in this article was supported by the Laboratory Directed Research and Development program of Los Alamos National Laboratory under project number 20180709PRD1.

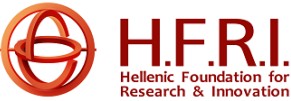

# A   Description of ancillary data file

In the ancillary data file of the arXiv submission, we give full results for a number of quantities, presented on the form $\langle quantity\rangle\langle fx\text{-}pt\rangle\langle expansion\rangle$, where $\langle fx\text{-}pt\rangle$ ranges over the fixed points ON, Chiral and Anti; and $\langle expansion\rangle$ over the expansions Eps and N. In Table 6 we list the options for $\langle quantity\rangle$.

The expressions make use of the constants

$$\texttt{sqRmn} = \sqrt{R_{mn}} = \sqrt{m^2 - 10mn + n^2 - 4m - 4n + 52}, \tag{81}$$

$$\texttt{eta1} = \eta_1^{O(N)} = \frac{(\mu - 2)\Gamma(2\mu - 1)}{\Gamma(\mu)\Gamma(\mu + 1)\pi\csc(\pi\mu)}, \tag{82}$$

implemented as sqRmnval and eta1val respectively (see (60)). The values that we present for deltaScalars$\langle fx\text{-}pt\rangle$Eps contain results from the literature at order $\varepsilon^2$, extracted from [25].

Data for ope$\langle fx\text{-}pt\rangle$Eps include long expressions that we do not include in the ancillary data file but can be made available upon request.

# B   Explicit formulas used in the main text

Here we give a few explicit formulas used in the main text. In (42) we have

$$a_{0,\ell}^{\text{GFF}} = \frac{2\Gamma(\ell + 1)^2}{\Gamma(2\ell + 1)}\Big[1 + (S_1(2\ell) - 2S_1(\ell))\left(\varepsilon - 2\gamma_\phi^{(2)}\varepsilon^2\right)$$
$$+ \frac{\varepsilon^2}{4}\left(8S_1(\ell)(S_1(\ell) - S_1(2\ell)) + 2S_1(2\ell)^2 - 3S_2(\ell) + 2S_2(2\ell)\right)\Big], \tag{83}$$

Table 6: Quantities given in the ancillary data file.

| Expression | Quantity | Order | Depends on |
|---|---|---|---|
| deltaPhi⟨fx-pt⟩Eps | $\Delta_\phi$ | $\varepsilon^3$ | $m, n, \varepsilon, $ sqRmn |
| deltaScalars⟨fx-pt⟩Eps | $\{\Delta_{\phi_R^2}\}_{R=1,\ldots,5}$ | $\varepsilon^2$ | $m, n, \varepsilon, $ sqRmn |
| deltaSpinning⟨fx-pt⟩Eps | $\{\Delta_{R,\ell}\}_{R=1,\ldots,9}$ | $\varepsilon^3$ | $\ell, m, n, \varepsilon, $ sqRmn |
| CT⟨fx-pt⟩Eps | $C_T/C_{T,\text{free}}$ | $\varepsilon^3$ | $m, n, \varepsilon, $ sqRmn |
| CJA⟨fx-pt⟩Eps | $C_{J_A}/C_{J_A,\text{free}}$ | $\varepsilon^3$ | $m, n, \varepsilon, $ sqRmn |
| CJB⟨fx-pt⟩Eps | $C_{J_B}/C_{J_B,\text{free}}$ | $\varepsilon^3$ | $m, n, \varepsilon, $ sqRmn |
| deltaPhi⟨fx-pt⟩N | $\Delta_\phi$ | $1/n$ | $m, n, \mu, $ eta1 |
| deltaScalars⟨fx-pt⟩N | $\{\Delta_R\}_{R=1,\ldots,5}$ | $1/n$ | $m, n, \mu, $ eta1 |
| deltaSpinning⟨fx-pt⟩N | $\{\Delta_{R,\ell}\}_{R=1,\ldots,9}$ | $1/n$ | $\ell, m, n, \mu, $ eta1 |
| CT⟨fx-pt⟩N | $C_T/C_{T,\text{free}}$ | $1/n$ | $m, n, \mu$ |
| CJA⟨fx-pt⟩N | $C_{J_A}/C_{J_A,\text{free}}$ | $1/n$ | $m, n, \mu$ |
| CJB⟨fx-pt⟩N | $C_{J_B}/C_{J_B,\text{free}}$ | $1/n$ | $m, n, \mu$ |
| ope⟨fx-pt⟩N | $\{a_{R,\ell}\}_{R=1,\ldots,9}$ | $1/n$ | $\ell, m, n, \mu, $ eta1 |

which is an expansion of the general formula for the OPE coefficients of generalized free fields [77],

$$
a_{n,\ell}^{\text{GFF}}\big|_{\Delta_\phi} = \frac{2(\Delta_\phi + 1 - \mu)_n^2 (\Delta_\phi)_{n+\ell}^2}{\ell!\, n!\, (\ell + \mu)_n (2\Delta_\phi + n + 1 - 2\mu)_n (2\Delta_\phi + \ell + n - \mu)_n (2\Delta_\phi + 2n + \ell - 1)_\ell}, \tag{84}
$$

where $(a)_n = \frac{\Gamma(a+n)}{\Gamma(a)}$ is the Pochhammer symbol.

In Inversion 4, $E_\pm$ is given by

$$
E_\pm = \alpha \kappa^2 \frac{2\bar{h} - 1}{4} \left[ \frac{2\pi \csc(\pi\mu)\big((\mu-2)(\mathbf{S_1}[\mu-1](\bar{h}) + \pi\cot(\pi\mu)) - 1\big) \pm 2\left(\frac{3-\mu}{\mu-2} - S_1(\mu-2)\right)}{J^2(\mu-2)^2} \right.
$$
$$
\left. \pm \frac{\mu(\mu-1)B(\mu,\bar{h}) + 2(S_2(\mu-2) - \zeta_2) + \frac{2(\mu-3)}{(\mu-2)^2}}{(J^2 - (\mu-1)(\mu-2))(\mu-2)} - \frac{2\pi\csc(\pi\mu)}{J^4(\mu-2)} \right], \tag{85}
$$

where

$$
B(\mu,\bar{h}) = \frac{{}_4F_3\left(\begin{matrix} 1,1,2,\mu+1 \\ 3,3-\bar{h},\bar{h}+2 \end{matrix}\,\bigg|\,1\right)}{J^2(J^2-2)} - \frac{2\pi\Gamma(\bar{h})\Gamma(\mu+\bar{h}-1)}{J^2\Gamma(\mu+1)\sin(\pi\bar{h})\Gamma(2\bar{h})}\,{}_3F_2\left(\begin{matrix} \bar{h}-1,\bar{h},\bar{h}+\mu-1 \\ 2\bar{h},\bar{h}+1 \end{matrix}\,\bigg|\,1\right), \tag{86}
$$

and in expressions (76) and (77) we have

$$
c_1 = \left(\frac{4+2\mu-\mu^2}{2\mu(2-\mu)} + \pi\cot(\pi\mu) + S_1(2\mu-2)\right)\frac{\eta_1^{O(N)}}{\mu+1},
$$
$$
c_2 = \frac{2\mu-1}{\mu(\mu-1)}\eta_1^{O(N)},
$$
$$
c_3 = \left(\frac{\mu^3 - 6\mu^2 + 11\mu - 4}{\mu(\mu-1)(2-\mu)} + \pi\cot(\pi\mu) + S_1(2\mu-3)\right)\eta_1^{O(N)}. \tag{87}
$$

# C  Crossing equations in $O(m) \times O(n)$ theories

For the single-correlator numerical bootstrap, i.e. that of the four-point function $\langle \phi_{ar}\phi_{bs}\phi_{ct}\phi_{du}\rangle$, we can easily work out the crossing equations by noticing that, when thinking of $\phi$ as a matrix,

$O(m)$ acts on the row index and $O(n)$ on the column index. Consequently, the projectors that allow for the decomposition of the four-point function into invariant subspaces can be formed as products of the well-known $O(N)$ projectors,

$$\hat{P}^S_{N;ijkl} = \frac{1}{N}\delta_{ij}\delta_{kl}, \quad \hat{P}^T_{N;ijkl} = \frac{1}{2}(\delta_{ik}\delta_{jl} + \delta_{il}\delta_{jk} - \frac{2}{N}\delta_{ij}\delta_{kl}), \quad \hat{P}^A_{N;ijkl} = \frac{1}{2}(\delta_{ik}\delta_{jl} - \delta_{il}\delta_{jk}),$$
(88)

where the subscript $N$ is introduced to capture the relation $\delta_{ij}\delta^{ij} = N$. The four-point function $\langle \phi_{ar}\phi_{bs}\phi_{ct}\phi_{du} \rangle$ can be decomposed as

$$\langle \phi_{ar}(x_1)\phi_{bs}(x_2)\phi_{ct}(x_3)\phi_{du}(x_4) \rangle = \frac{1}{(x_{12}^2 x_{34}^2)^{\Delta_\phi}} \sum_R \sum_{\mathcal{O}_R} c^2_{\mathcal{O}_R} P^R_{arbsctdu} G_{\Delta_R,\ell_R}(u,v),$$
(89)

where the sum over $R$ runs over the representations $S, W, X, Y, Z, A, B, C, D$, $x_{ij} = x_i - x_j$, $c^2_{\mathcal{O}_R}$ are squared OPE coefficients and $G_{\Delta_R,\ell_R}(u,v)$ are conformal blocks[19] that are functions of the conformally-invariant cross-ratios defined in (12). The projectors in (89) are

$$P^S_{arbsctdu} = \hat{P}^S_{m;abcd}\hat{P}^S_{n;rstu}, \quad P^W_{arbsctdu} = \hat{P}^T_{m;abcd}\hat{P}^S_{n;rstu}, \quad P^X_{arbsctdu} = \hat{P}^S_{m;abcd}\hat{P}^T_{n;rstu},$$
$$P^Y_{arbsctdu} = \hat{P}^T_{m;abcd}\hat{P}^T_{n;rstu}, \quad P^Z_{arbsctdu} = \hat{P}^A_{m;abcd}\hat{P}^A_{n;rstu}, \quad P^A_{arbsctdu} = \hat{P}^A_{m;abcd}\hat{P}^S_{n;rstu},$$
$$P^B_{arbsctdu} = \hat{P}^S_{m;abcd}\hat{P}^A_{n;rstu}, \quad P^C_{arbsctdu} = \hat{P}^A_{m;abcd}\hat{P}^T_{n;rstu}, \quad P^D_{arbsctdu} = \hat{P}^T_{m;abcd}\hat{P}^A_{n;rstu}.$$
(90)

For the crossing equation we define

$$F^{\pm}_{\Delta,\ell}(u,v) = v^{\Delta_\phi} g_{\Delta,\ell}(u,v) \pm u^{\Delta_\phi} g_{\Delta,\ell}(v,u),$$
(91)

and we find[20]

$$\sum_{S^+} c^2_{\mathcal{O}} \begin{pmatrix} F^-_{\Delta,\ell} \\ 0 \\ 0 \\ 0 \\ 0 \\ F^+_{\Delta,\ell} \\ 0 \\ 0 \\ 0 \end{pmatrix} + \sum_{W^+} c^2_{\mathcal{O}} \begin{pmatrix} -\frac{2}{m}F^-_{\Delta,\ell} \\ 0 \\ F^-_{\Delta,\ell} \\ F^-_{\Delta,\ell} \\ 0 \\ -\frac{2}{m}F^+_{\Delta,\ell} \\ 0 \\ -F^+_{\Delta,\ell} \\ F^+_{\Delta,\ell} \end{pmatrix} + \sum_{X^+} c^2_{\mathcal{O}} \begin{pmatrix} -\frac{2}{n}F^-_{\Delta,\ell} \\ F^-_{\Delta,\ell} \\ F^-_{\Delta,\ell} \\ 0 \\ 0 \\ -\frac{2}{n}F^+_{\Delta,\ell} \\ F^+_{\Delta,\ell} \\ F^+_{\Delta,\ell} \\ 0 \end{pmatrix} + \sum_{Y^+} c^2_{\mathcal{O}} \begin{pmatrix} (1+\frac{4}{mn})F^-_{\Delta,\ell} \\ (1-\frac{2}{m})F^-_{\Delta,\ell} \\ -2(\frac{1}{m}+\frac{1}{n})F^-_{\Delta,\ell} \\ (1-\frac{2}{n})F^-_{\Delta,\ell} \\ 2F^-_{\Delta,\ell} \\ -(1-\frac{4}{mn})F^+_{\Delta,\ell} \\ -(1+\frac{2}{m})F^+_{\Delta,\ell} \\ -2(\frac{1}{m}-\frac{1}{n})F^+_{\Delta,\ell} \\ -(1+\frac{2}{n})F^+_{\Delta,\ell} \end{pmatrix}$$

$$+ \sum_{Z^+} c^2_{\mathcal{O}} \begin{pmatrix} F^-_{\Delta,\ell} \\ -F^-_{\Delta,\ell} \\ 0 \\ -F^-_{\Delta,\ell} \\ 2F^-_{\Delta,\ell} \\ -F^+_{\Delta,\ell} \\ F^+_{\Delta,\ell} \\ 0 \\ F^+_{\Delta,\ell} \end{pmatrix} + \sum_{A^-} c^2_{\mathcal{O}} \begin{pmatrix} 0 \\ 0 \\ -F^-_{\Delta,\ell} \\ F^-_{\Delta,\ell} \\ 0 \\ 0 \\ 0 \\ F^+_{\Delta,\ell} \\ F^+_{\Delta,\ell} \end{pmatrix} + \sum_{B^-} c^2_{\mathcal{O}} \begin{pmatrix} 0 \\ F^-_{\Delta,\ell} \\ -F^-_{\Delta,\ell} \\ 0 \\ 0 \\ 0 \\ F^+_{\Delta,\ell} \\ -F^+_{\Delta,\ell} \\ 0 \end{pmatrix}$$

---

[19]In our numerical bootstrap considerations we define conformal blocks using the conventions of [70].

[20]In (92) we suppress the labeling of the $F_{\Delta,\ell}$'s and $c^2_{\mathcal{O}}$'s with the appropriate index $I$. The appropriate labeling, however, is obvious from the overall sum in each term.

$$+ \sum_{C^-} c_{\mathcal{O}}^2 \begin{pmatrix} -F_{\Delta,\ell}^- \\ -F_{\Delta,\ell}^- \\ \frac{2}{n}F_{\Delta,\ell}^- \\ (1-\frac{2}{n})F_{\Delta,\ell}^- \\ 2F_{\Delta,\ell}^- \\ F_{\Delta,\ell}^+ \\ F_{\Delta,\ell}^+ \\ -\frac{2}{n}F_{\Delta,\ell}^+ \\ -(1+\frac{2}{n})F_{\Delta,\ell}^+ \end{pmatrix} + \sum_{D^-} c_{\mathcal{O}}^2 \begin{pmatrix} -F_{\Delta,\ell}^- \\ (1-\frac{2}{m})F_{\Delta,\ell}^- \\ \frac{2}{m}F_{\Delta,\ell}^- \\ -F_{\Delta,\ell}^- \\ 2F_{\Delta,\ell}^- \\ F_{\Delta,\ell}^+ \\ -(1+\frac{2}{m})F_{\Delta,\ell}^+ \\ \frac{2}{m}F_{\Delta,\ell}^+ \\ F_{\Delta,\ell}^+ \end{pmatrix} = \begin{pmatrix} 0 \\ 0 \\ 0 \\ 0 \\ 0 \\ 0 \\ 0 \\ 0 \\ 0 \end{pmatrix}. \quad (92)$$

The signs that appear as superscripts in the various irrep symbols indicate the spins of the operators we sum over in the corresponding term: even when positive and odd when negative.

When $m = n$ there is a reduction in the number of crossing equations. In this case, instead of separate projectors $P^W_{arbsctdu}$ and $P^X_{arbsctdu}$, we only have the projector $P^{WX}_{arbsctdu} = P^W_{arbsctdu} + P^X_{arbsctdu}$, and similarly $P^{AB}_{arbsctdu} = P^A_{arbsctdu} + P^B_{arbsctdu}$ and $P^{CD}_{arbsctdu} = P^C_{arbsctdu} + P^D_{arbsctdu}$, always with $m = n$. The crossing equation in the $m = n$ case is

$$\sum_{S^+} c_{\mathcal{O}}^2 \begin{pmatrix} F_{\Delta,\ell}^- \\ 0 \\ 0 \\ 0 \\ F_{\Delta,\ell}^+ \\ 0 \end{pmatrix} + \sum_{WX^+} c_{\mathcal{O}}^2 \begin{pmatrix} -\frac{4}{n}F_{\Delta,\ell}^- \\ F_{\Delta,\ell}^- \\ F_{\Delta,\ell}^- \\ 0 \\ -\frac{4}{n}F_{\Delta,\ell}^+ \\ F_{\Delta,\ell}^+ \end{pmatrix} + \sum_{Y^+} c_{\mathcal{O}}^2 \begin{pmatrix} (1+\frac{4}{n^2})F_{\Delta,\ell}^- \\ (1-\frac{2}{n})F_{\Delta,\ell}^- \\ -\frac{2}{n}F_{\Delta,\ell}^- \\ F_{\Delta,\ell}^- \\ -(1-\frac{4}{n^2})F_{\Delta,\ell}^+ \\ -(1+\frac{2}{n})F_{\Delta,\ell}^+ \end{pmatrix} + \sum_{Z^+} c_{\mathcal{O}}^2 \begin{pmatrix} F_{\Delta,\ell}^- \\ -F_{\Delta,\ell}^- \\ 0 \\ F_{\Delta,\ell}^- \\ -F_{\Delta,\ell}^+ \\ F_{\Delta,\ell}^+ \end{pmatrix}$$

$$+ \sum_{AB^-} c_{\mathcal{O}}^2 \begin{pmatrix} 0 \\ F_{\Delta,\ell}^- \\ -F_{\Delta,\ell}^- \\ 0 \\ 0 \\ F_{\Delta,\ell}^+ \end{pmatrix} + \sum_{CD^-} c_{\mathcal{O}}^2 \begin{pmatrix} -F_{\Delta,\ell}^- \\ -\frac{1}{n}F_{\Delta,\ell}^- \\ \frac{1}{n}F_{\Delta,\ell}^- \\ F_{\Delta,\ell}^- \\ F_{\Delta,\ell}^+ \\ -\frac{1}{n}F_{\Delta,\ell}^+ \end{pmatrix} = \begin{pmatrix} 0 \\ 0 \\ 0 \\ 0 \\ 0 \\ 0 \end{pmatrix}.$$

$$(93)$$

When $n = 2$, (93) is equivalent to the crossing equation derived for the global symmetry group $O(2)^2 \rtimes S_2$ in [39, Eq. (2.15)]. The representations $S, WX, Y, Z, AB, CD$ in (93) correspond to the representations $S, Z, Y, X, A, B$, respectively, in [39, Eq. (2.15)]. Theories with $O(2)^2/\mathbb{Z}_2$ and $O(2)^2 \rtimes S_2$ global symmetry are equivalent at the Lagrangian level in $d = 4-\varepsilon$ dimensions [14,25]. The groups $O(2)^2/\mathbb{Z}_2$ and $O(2)^2 \rtimes S_2$ each have a fully-symmetric four-index invariant tensor, as well as an additional four-index invariant tensor whose symmetry properties are such that it does not generate quartic invariants in the corresponding Lagrangians. For the $O(2)^2/\mathbb{Z}_2$ case the relevant invariant tensor was called $w_2$ in [25, Eq. (5.81)], and for $O(2)^2 \rtimes S_2$ it was called $w$ in [25, Eq. (5.96)]. The relevance of the $O(2)^2/\mathbb{Z}_2$ theory for experiments in stacked triangular antiferromagnets, helimagnets and structural phase transitions has been discussed extensively in [39] and references therein.

In this paper we also consider a mixed correlator bootstrap involving the four-point functions $\langle \phi\phi\phi\phi \rangle$, $\langle \phi\phi SS \rangle$ and $\langle SSSS \rangle$, where $S$ is the first scalar singlet in the $\phi \times \phi$ OPE. The crossing equations for this system are straightforward to derive due to the fact that $S$ transforms in the singlet representation, and so in the OPE $\phi \times S$ we find only operators that transform in $\phi$'s representation under the global symmetry.

As usual, our numerical treatment involves two steps, namely generation of an xml file that encodes the problem and subsequently its solution with a numerical algorithm. For the first step we use PyCFTBoot [70], and for the second SDPB [78]. For the single correlator bootstrap we use the numerical parameters m_max = 8, n_max = 11, k_max = 42 in PyCFTBoot, and we include spins up to l_max = 36. For the mixed correlator bootstrap we use the numerical parameters m_max = 7, n_max = 9, k_max = 40 in PyCFTBoot, and we include spins up to l_max = 30. The binary precision for the xml files in both cases is 660 digits. SDPB is run with the options -precision=860, -findPrimalFeasible, -findDualFeasible, -primalErrorThreshold=1e-30 and finally -dualErrorThreshold=1e-15. Default values are used for other parameters of the solver.

# D  Mellin bootstrap for any global symmetry

In this appendix we will revisit some equations from the analytic boostrap in Mellin space and apply them to the $\varepsilon$ expansion for $\phi^4$ theories with arbitrary global symmetry. The framework is described in detail in [79] and was generalized to global symmetry in [38], focusing on $O(N)$ and hypercubic symmetry. We will show how the Mellin space bootstrap reproduces the system of equations (38), which we used to find the perturbative fixed points for a given global symmetry. In addition, we will show that

$$c^2_{\phi\phi\phi^2_R} = a_{R,0}(1 - g_R\varepsilon) + O(\varepsilon^2), \tag{94}$$

i.e. that $\alpha_R = -g_R$ in (45), a result needed for our computations in section 3.2 at order $\varepsilon^3$.

The Mellin space bootstrap considers manifestly crossing symmetric expressions in the Mellin space variables $s, t$, equivalent to sums over Witten diagrams for exchanges of operators parametrized by $\Delta, \ell$. Consistency with the OPE implies equations derived from the cancellation of spurious poles. For poles generated by the Mellin variable $s$ one gets the equation

$$\sum_{\Delta,\ell} \left( c^{ijkl(s)}_{\Delta,\ell} M^{(s)}_{\Delta,\ell}(s,t) + c^{ijkl(t)}_{\Delta,\ell} M^{(t)}_{\Delta,\ell}(s,t) + c^{ijkl(u)}_{\Delta,\ell} M^{(u)}_{\Delta,\ell}(s,t) \right)\Big|_{s=\Delta_\phi} = 0, \tag{95}$$

valid for all $t$, where we have generalized the notation of [79] by adding global symmetry indices. From (95), one derives the *constraint equations*, by projecting to a given spin $\ell$ in the $s$-channel of the $R$ representation, using the fact that the $M_{\Delta,\ell}(s,t)$ can be expressed in terms of continuous Hahn polynomials. The first term becomes simply $\sum_\Delta c^R_{\Delta,\ell}$, but for the other two terms, operators of any spin $\ell'$ contribute. We arrive at constraint equations of the form

$$\sum_\Delta c^R_{\Delta,\ell} q^{R(2,s)}_{\Delta,\ell} + 2\sum_{\widetilde{R}} M_{R\widetilde{R}} \sum_{\Delta',\ell'} c^{\widetilde{R}}_{\Delta',\ell'} q^{\widetilde{R}(2,t)}_{\Delta',\ell|\ell'} = 0, \tag{96}$$

$$\sum_\Delta c^R_{\Delta,\ell} q^{R(1,s)}_{\Delta,\ell} + 2M_{RS} q^{(1,t)}_{\mathbb{1}} + 2\sum_{\widetilde{R}} M_{R\widetilde{R}} \sum_{\Delta',\ell'} c^{\widetilde{R}}_{\Delta',\ell'} q^{\widetilde{R}(1,t)}_{\Delta',\ell|\ell'} = 0, \tag{97}$$

where, again, the exact form of the involved quantities can be found in [79]. We have collected the $t$ and $u$ channel contributions under the label $t$, and the appearance of the matrix $M_{R\widetilde{R}}$ follows from using the projectors of (11). Equations (96) and (97) are generalizations of equations (2.39)–(2.44) of [38]. We will evaluate them for $\ell = 0$, assuming that $\Delta_{\phi^2_R} = 2 - \varepsilon + g_R\varepsilon + O(\varepsilon^2)$ and $c^2_{\phi\phi\phi^2_R} = a_{R,0}(1 + \alpha_R\varepsilon) + O(\varepsilon^2)$, and expanding to order $\varepsilon$. Only $\ell' = 0$ contributes to the sum, and following [79] we substitute

$$c^R_{\Delta,0} q^{R(2,s)}_{\Delta,0} = -a_{R,0} g_R(g_R - 1)\frac{\varepsilon}{2} + O(\varepsilon^2), \tag{98}$$

$$c_{\Delta,0}^{R} q_{\Delta,0}^{R(1,s)} = a_{R,0} \left( 1 + (\alpha_R + g_R - 1 - \gamma_E)\varepsilon \right) + O(\varepsilon^2), \tag{99}$$

$$c_{\Delta',0'}^{\widetilde{R}} q_{\Delta',0|0'}^{\widetilde{R}(2,t)} = -a_{\widetilde{R},0}\, g_{\widetilde{R}}^2 \frac{\varepsilon}{2} + O(\varepsilon^2), \tag{100}$$

$$c_{\Delta',0'}^{\widetilde{R}} q_{\Delta',0|0'}^{\widetilde{R}(1,t)} = 0 + O(\varepsilon^2), \tag{101}$$

$$q_{\mathbb{1}}^{(1,t)} = -1 + (1 + \gamma_E)\varepsilon + O(\varepsilon^2), \tag{102}$$

where $\gamma_E$ is the Euler–Mascheroni constant.

With these substitutions we can solve the constraint equations. From (97) at order $\varepsilon^0$ we get $a_{R,0} = 2M_{RS}$, in agreement with (31) above. Feeding this into (96) we get

$$-M_{RS}\, g_R(g_R - 1)\varepsilon - 2\sum_{\widetilde{R}} M_{R\widetilde{R}} M_{\widetilde{R}S}\, g_{\widetilde{R}}^2 \varepsilon = 0 + O(\varepsilon^2), \tag{103}$$

which exactly agrees with (38). Finally, by looking at (97) at order $\varepsilon$ we get $\alpha_R + g_R = 0$, proving (94).

In [38], the CFT-data was computed to order $\varepsilon^3$ for $O(N)$ and hypercubic symmetry, and we believe that this will generalize to arbitrary global symmetry. From such implementation, one can derive the order $\varepsilon^3$ correction to the OPE coefficient $c_{\phi\phi\phi_R^2}^2$ (taking $\gamma_{\phi_R^2}^{(3)}$ as input), an observable that is inaccessible from large spin perturbation theory in its present formulation.

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
