# Peer review of "Analytic and Numerical Bootstrap of CFTs with $O(m)\times O(n)$ Global Symmetry in 3D"

_SciPost Physics, doi:SciPost Phys. 9, 035 (2020)_

## Round 2 · Referee Report · Anonymous (Referee 1) · 2020-7-26

Strengths

1) Detailed discussion of the results. 2) The presentation is very balanced: the reliability of the results is always discussed clearly and honestly.

Weaknesses

1) Too long. It would have been probably better to have different papers for the large N behavior and for the O(2)timesO(2) and O(2)times O(3) behavior

Report

I think the paper should be published. The paper studies a class of models that have been of great interest in the last 30 years. Nonetheless, the critical
behavior in the physically interesting cases is still debated. Thus, I have no reservations on its significance. Moreover, the paper is clearly written (I have a few comments to improve the presentation).

Requested changes

Given the length of the paper it would be interesting to summarize the results for the two physically interesting cases O(2)timesO(3) and O(2)timesO(2) in a table in the conclusions. One should report the results using the standard notation using the in statistical physics (exponents nu, eta, omega, crossover exponents phi), comparing them with the literature results . This would allow workers in statistical and condensed matter to grab the results without the need of looking inside the full article and of understanding the notations used in the conformal bootstrap method.

---

## Round 2 · Referee Report · Anonymous (Referee 2) · 2020-8-12

Strengths

  • contains several details which make possible to reproduce most of the analytic results without much work
  • mixed correlator approach
  • synergy between analytic and numerical approaches

Weaknesses

  • not clearly specified the results, in some places they appear to be hidden by technical details

Report

The paper is overall clearly written and contains several interesting results. I have a list of suggestions/questions that I would like the authors to address before the paper can be accepted for publication by SciPost.

Requested changes

1-in the last paragraph before 1.1, it is mentioned that m is small and n is large. Is it enough that m is smaller (even much smaller) than m or not? 2- there are two expansions made, the small $\epsilon$ and the large $n$ expansion. Specially in the introduction, it is not clear which is the order of the expansions. Is there any issue related to it? 3-regarding the non unitarity: how would you have expected to see it? As it is pointed out in the paper, the numerical bootstrap only finds bounds or islands of theories which are unitary so it is not clear how the non unitarity would be signalled both in the numerical and analytic approaches.

---

## Round 3 · Author Response

We would like to thank both referees for their comments on our manuscript. A list of changes to address the referees' comments can be found below.

---

## Round 3 · List of Changes

1. We added tables with results for critical exponents in our conclusion section, as suggested in Report 1. Hopefully this also ameliorates the indicated weakness in Report 2.
2. In response to Requested Change 1 in Report 2, we emphasize that our expansion parameter is 1/n, not m/n. This is clearly stated in the paragraph before subsection 1.2.
3. We added footnote 1 to address Requested Change 2 in Report 2.
4. We added footnote 17 to address Requested Change 3 in Report 2.

You are currently on this page

Resubmission 2004.14388v3 on 26 August 2020

---

## Editorial Decision

published